# Viscoelastic properties of suspended cells measured with shear flow deformation cytometry

Richard Gerum[1,2], Elham Mirzahossein[1], Mar Eroles[3], Jennifer Elsterer[1], Astrid Mainka[1], Andreas Bauer[1], Selina Sonntag[1], Alexander Winterl[1], Johannes Bartl[1], Lena Fischer[1], Shada Abuhattum[4], Ruchi Goswami[4], Salvatore Girardo[4], Jochen Guck[1,4], Stefan Schrüfer[5], Nadine Ströhlein[1], Mojtaba Nosratlo[1], Harald Herrmann[6], Dorothea Schultheis[6], Felix Rico[3], Sebastian Johannes Müller[7], Stephan Gekle[7], Ben Fabry[1]*

[1]Department of Physics, Friedrich-Alexander University Erlangen-Nurnberg, Erlangen, Germany; [2]Department of Physics and Astronomy, York-University Toronto, Ontario, Canada; [3]Aix-Marseille Universite´, CNRS, Inserm, LAI, Turing centre for living systems, Marseille, France; [4]Max Planck Institute for the Science of Light and Max-Planck-Zentrum fur Physik und Medizin, Erlangen, Germany; [5]Institute of Polymer Materials, Friedrich-Alexander University Erlangen-Nurnberg, Erlangen, Germany; [6]Institute of Neuropathology, University Hospital Erlangen, Erlangen, Germany; [7]Department of Physics, University of Bayreuth, Bayreuth, Germany

*For correspondence:
ben.fabry@fau.de

**Abstract** Numerous cell functions are accompanied by phenotypic changes in viscoelastic properties, and measuring them can help elucidate higher level cellular functions in health and disease. We present a high-throughput, simple and low-cost microfluidic method for quantitatively measuring the elastic (storage) and viscous (loss) modulus of individual cells. Cells are suspended in a high-viscosity fluid and are pumped with high pressure through a 5.8 cm long and 200 μm wide microfluidic channel. The fluid shear stress induces large, ear ellipsoidal cell deformations. In addition, the flow profile in the channel causes the cells to rotate in a tank-treading manner. From the cell deformation and tank treading frequency, we extract the frequency-dependent viscoelastic cell properties based on a theoretical framework developed by R. Roscoe [1] that describes the deformation of a viscoelastic sphere in a viscous fluid under steady laminar flow. We confirm the accuracy of the method using atomic force microscopy-calibrated polyacrylamide beads and cells. Our measurements demonstrate that suspended cells exhibit power-law, soft glassy rheological behavior that is cell-cycle-dependent and mediated by the physical interplay between the actin filament and intermediate filament networks.

## Editor's evaluation

This paper describes an inexpensive but very powerful microfluidic approach to quantitatively determine the viscoelastic properties of living cells from their deformation in a flow. Its implementation seems simple so that even people not specialized in cell mechanics can use it, and the method offers the possibility to perform measurements on a large number of cells (up to 50-100 per second). The data are compelling and this technique should set a new standard in the field.

**eLife digest** Cells in the human body are viscoelastic: they have some of the properties of an elastic solid, like rubber, as well as properties of a viscous fluid, like oil. To carry out mechanical tasks – such as, migrating through tissues to heal a wound or to fight inflammation – cells need the right balance of viscosity and elasticity. Measuring these two properties can therefore help researchers to understand important cell tasks and how they are impacted by disease.

However, quantifying these viscous and elastic properties is tricky, as both depend on the time-scale they are measured: when pressed slowly, cells appear soft and liquid, but they turn hard and thick when rapidly pressed. Here, Gerum et al. have developed a new system for measuring the viscosity and elasticity of individual cells that is fast, simple, and inexpensive.

In this new method, cells are suspended in a specialized solution with a consistency similar to machine oil which is then pushed with high pressure through channels less than half a millimeter wide. The resulting flow of fluid shears the cells, causing them to elongate and rotate, which is captured using a fast camera that takes 500 images per second. Gerum et al. then used artificial intelligence to extract each cell's shape and rotation speed from these images, and calculated their viscosity and elasticity based on existing theories of how viscoelastic objects behave in fluids.

Gerum et al. also investigated how the elasticity and viscosity of cells changed with higher rotation frequencies, which corresponds to shorter time-scales. This revealed that while higher frequencies made the cells appear more viscous and elastic, the ratio between these two properties remained the same. This means that researchers can compare results obtained from different experimental techniques, even if the measurements were carried out at completely different frequencies or time-scales.

The method developed by Gerum et al. provides a fast an inexpensive way for analyzing the viscosity and elasticity of cells. It could also be a useful tool for screening the effects of drugs, or as a diagnostic tool to detect diseases that affect the mechanical properties of cells.

## Introduction

Eukaryotic cells can carry out complex mechanical tasks such as cell division, adhesion, migration, invasion, and force generation. These mechanical activities in turn are essential for higher order cell functions including differentiation, morphogenesis, wound healing, or inflammatory responses. Since cell mechanical activities are accompanied by phenotypic changes in the cell's viscoelastic properties, measuring them can help elucidate higher order cell functions in health and disease (*Urbanska et al., 2020*). For example, the activation of neutrophils in response to pro-inflammatory agents is typically accompanied by a marked increase in cell stiffness (*Frank, 1990*; *Fabry et al., 2001*), which can subsequently lead to a sequestration of the stiffened cells in small capillaries for example of the lung (*Doerschuk et al., 1993*). This process may be relevant for the progression and exacerbation of inflammatory diseases such as coronavirus disease 2019.

In this report, we describe a quantitative, low-cost, high-throughput, and simple method to measure the viscoelastic properties of cells, specifically the storage modulus $G'$, and the loss modulus $G''$. The cells are suspended in a high-viscosity (0.5–10 Pa s) fluid (e.g. a 2% alginate solution) and are pumped at pressures of typically between 50 and 300 kPa through a several centimeter long microfluidic channel with a square cross section (200x200 µm in our set-up). The fluid shear stress induces large cell deformations that are imaged using a complementary metal-oxide-semiconductor (CMOS) camera at frame rates of up to 500 frames/s to achieve a measurement throughput of up to 100 cells/s. Images are stored and analyzed off-line at a speed of around 50 frames/s on a standard desktop PC equipped with a graphics card.

The method takes advantage of two physical principles: First, the shear stress profile inside a long microfluidic channel depends only on the pressure gradient along the channel, which can be precisely controlled, and the channel geometry, which is fixed. Importantly, the shear stress profile does not depend on the viscosity of the cell suspension medium and smoothly increases from zero at the channel center to a maximum value at the channel walls. Accordingly, cells appear circular near the channel center and become increasingly elongated near the channel walls. As the width of the channel is significantly larger than the cell diameter, fluid shear stresses remain approximately constant across the cell surface, which considerably simplifies the fluid dynamics computations compared to existing

microfluidic methods. From the stress-strain relationship, we estimate the storage modulus of the cell, which characterizes its elastic behavior.

Second, depending on the flow speed profile inside the channel, the cells rotate in a tank-treading manner, similar to a ball that is compressed between two counter-moving parallel plates. Shear-flow induced tank-treading was first theoretically explored by *Einstein, 1906*, and was later experimentally observed by H. Schmid-Schönbein et al. in sheared red blood cell suspensions (*Schmid-Schönbein and Wells, 1969*; *Fischer et al., 1978*). Tank-treading arises as the flow speed of the suspension fluid in contact with the cell surface facing the channel center is larger compared to the flow speed at the opposite side. Hence, the rotational speed of this tank-treading motion increases with increasing shear rate near the channel walls. Tank-treading in combination with the cell's viscous properties leads to energy dissipation, which limits the increase of cell strain at higher stresses near the channel walls. From this behavior, we extract the loss modulus of the cell, which characterizes its viscous behavior. Since the microfluidic channel is several centimeters long, most cells, with the exception of those in the center of the channel, have already undergone several full rotations before reaching the field of view. Therefore, the cell deformations are in a steady state, which is another major difference compared to existing microfluidic approaches and greatly simplifies the calculation of viscoelastic cell parameters.

For the calculation of viscoelastic cell parameter, we use a theoretical framework developed by *Roscoe, 1967* that describes the deformation of a viscoelastic sphere in a viscous fluid under steady shear flow. This theory allows us to compute the stiffness (shear modulus) and viscosity of a cell from 5 measurable parameters. First, the fluid shear stress acting on the cell must be known, which we compute based on the extension of Poiseuille's equation to channels with square cross-section (*Delplace, 2018*). Second, we measure the cell deformation (cell strain) from bright-field microscopy images. Third, we measure the alignment angle of the deformed cell with respect to the flow direction. This alignment angle depends on the ratio between cell viscosity and the viscosity of the suspension fluid. Fourth, we compute the local viscosity of the suspension fluid based on measurements of the radial flow speed profile in the channel, which we obtain from multiple images of the same cell during its passage through the channel. Fifth, since cell stiffness and cell viscosity are frequency-dependent, we measure the tank-treading frequency of each cell.

The Roscoe model assumes that cells behave as a Kelvin-Voigt body consisting of an elastic spring in parallel with a resistive (or viscous) dash-pot element. This then gives rise to a complex shear modulus with storage modulus $G'$ and loss modulus $G''$, measured at twice the tank treading frequency (because a given volume element inside the cell is compressed and elongated twice during a full rotation). Roscoe theory, however, makes no assumptions about how $G'$ and $G''$ might change as a function of frequency. A commonly used simplified assumption is that the elastic and viscous elements of the Kelvin-Voigt body are constant (*Fregin et al., 2019*). Hence, $G'$ plotted versus frequency would be flat, and $G''$ would increase proportional with frequency. An alternative and, as we will show in this report, a more accurate model, known as the structural damping formalism, predicts that both $G'$ and $G''$ increase with frequency according to a power-law (*Fabry et al., 2001*). In either case, to compare the stiffness and viscosity of cells that have experienced different tank-treading frequencies, it is important to scale the stiffness and fluidity of each cell to a reference frequency, for example of 1 Hz.

Using cell lines and calibrated polyacrylamide beads, we verify that our method provides accurate quantitative measurements of viscoelastic properties. Measurement results are not or only marginally influenced by experimental details such as the viscosity of the suspension fluid or the time point after suspending the cells. We demonstrate that the cell's viscoelastic properties measured with our method conform to soft glassy power-law rheology that has been reported for a wide range of cells measured with different methods. We also show that our method can be used for dose-response measurements of drugs that induce actin cytoskeleton disassembly, and that these responses are modulated by the cell cycle and the intermediate filament network of the cells.

## Results

### Measurement setup

We image the cells in bright-field mode while they are moving through the microchannel (*Figure 1a–c*). Using a neural network, we detect cells that are in focus at the mid-plane of the microchannel (*Figure 1b*), and segment their shapes (*Figure 1d*). We then quantify the cell position and cell shape

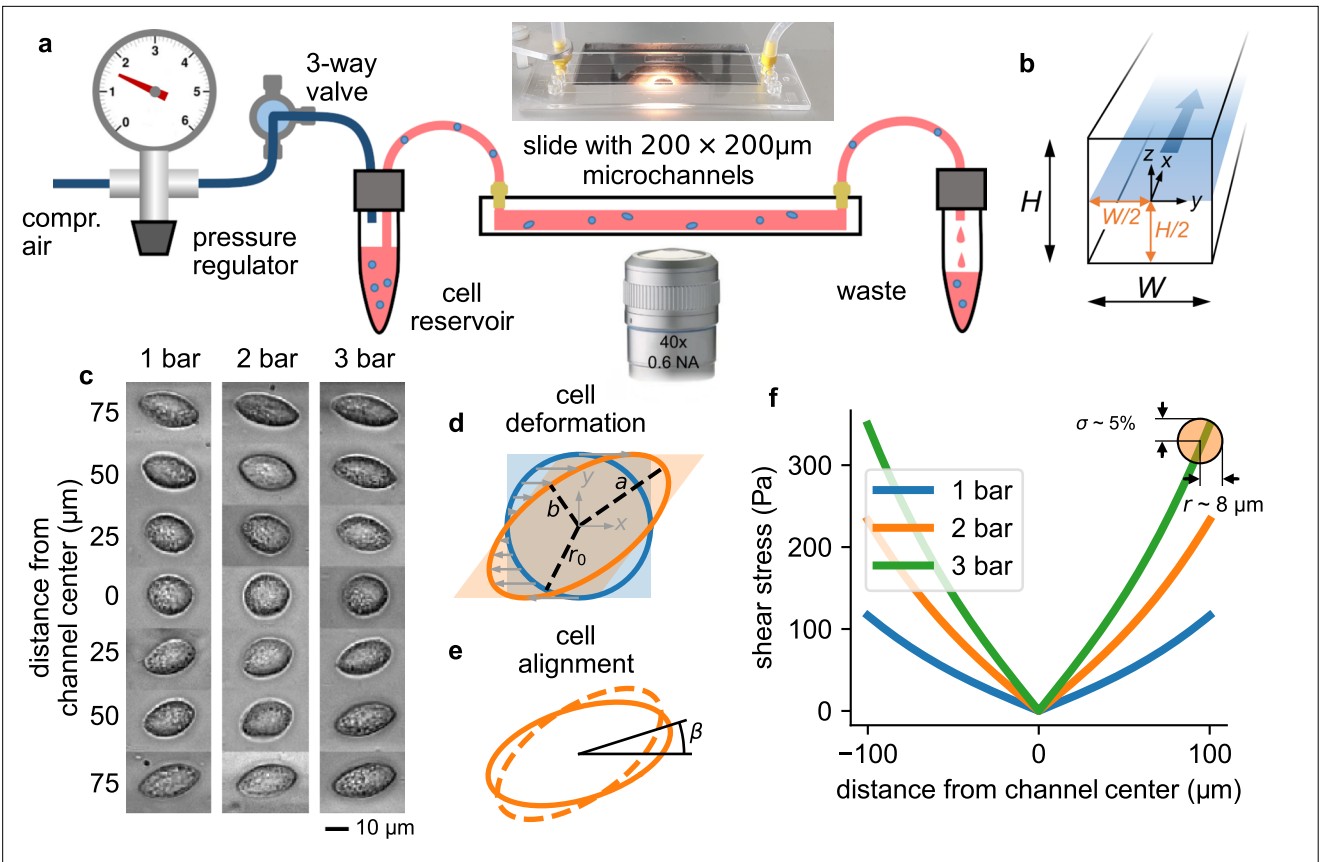

**Figure 1.** Measurement setup and principle. a, Schematic of the microfluidic device. b, Cross section through the microchannel with dimensions $W = H$ = 200 μm. The focal plane of the microscope at a height of $H/2$=100 μm is indicated by the blue shaded area. Fluid flow is in $x$ direction. c, Bright field images of NIH-3T3 cells under control conditions at different y-positions in a microchannel at a pressure of 1, 2, and 3 bar. Cells appear round in the channel center and become more elongated near the walls. d, Illustration of cell deformations under fluid shear. The circular cell with radius $r_0$ (blue) is transformed to an elliptical shape (orange) with semi-major axis $a$ and semi-minor axis $b$ depending on the ratio of fluid shear stress and the cell's shear modulus (*Equation 16*). e, The sheared cell (dashed outline) will partially align in flow direction (solid outline), characterized by an alignment angle β. This angle depends on the ratio of cell viscosity and suspension fluid viscosity (*Equation 17*). $a$, $b$, and β are measured from the segmented cell shapes. f, Fluid shear stress (computed according to *Equation 4*) versus distance from the channel center in $y$-direction for three different pressures of 1, 2, and 3 bar. Close to the channel wall, the shear stress varies by 5% across the cell surface for a typical cell with a radius of 8 μm (indicated by the orange circle). Cells that extend beyond the channel center are excluded form further analysis.

The online version of this article includes the following figure supplement(s) for figure 1:

**Figure supplement 1.** Software flow-chart.

by fitting an ellipse to the segmented cell image, from which we obtain the centroid coordinate ($x_0$, $y_0$), the length of the semi-major axis $a$ and the semi-minor axis $b$, and the angular orientation β of the major axis with respect to the $x$-(flow) direction (*Figure 1e*). From $a$ and $b$, we compute the cell strain $\epsilon$ using Equation 10 (*Figure 2a*). We also compute the local fluid shear stress $\sigma(y_0)$ for a cell-free fluid at the cell's centroid position using *Equation 4* (*Figure 1f*).

## Cell deformations under fluid shear stress

Cells are nearly circular in the center, and they elongate and align in flow direction near the channel walls (*Figure 1c*, *Figure 2a, b*) where they are exposed to higher fluid shear stress (*Figure 1f*). Cells imaged at the same position within the channel also tend to become more elongated with increasing pressure (*Figure 1c*). When we plot cell strain $\epsilon$ versus shear stress σ across the microfluidic channel (*Figure 2c*), we find that the cell strain increases non-linearly with increasing fluid shear stress. In particular, the slope of the strain versus stress relationship decreases for higher stress values. This behavior is predominantly due to a dissipative process caused by the tank tread-like motion of the cells.

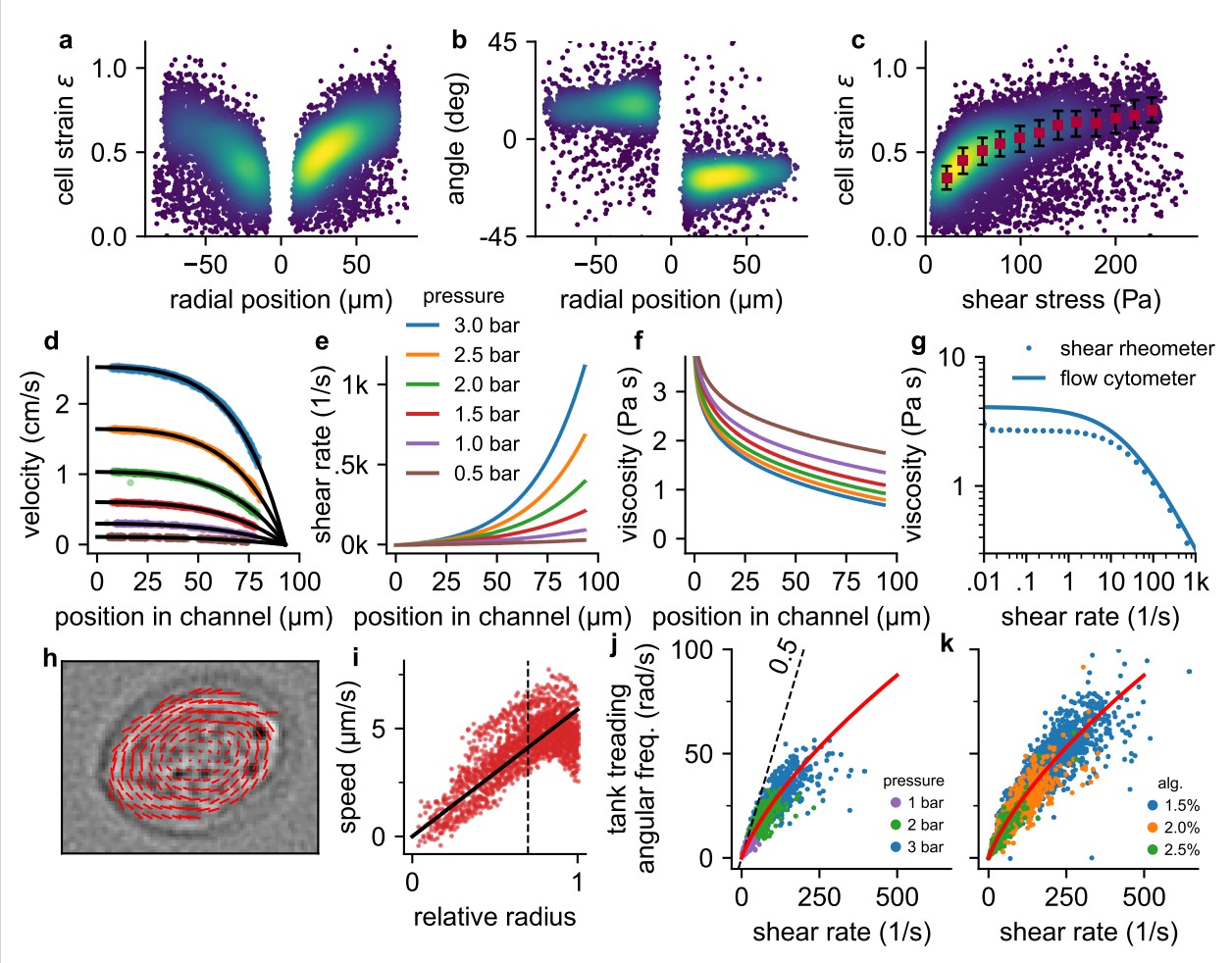

**Figure 2.** Cell responses to shear stress and shear rate. a, Cell strain versus radial ($y$) position in the channel for NIH-3T3 cells at a pressure of 3 bar. Each data point corresponds to a single cell. Colors indicate Gaussian kernel density. b, Cell alignment angle β versus radial position in the channel ($y$) for the same cells as in a. c, Cell strain versus shear stress for the same cells as in a. Red squares indicate median values over shear stress bins of 20 Pa starting from 10 Pa, error bars indicate quartiles. d, Fluid flow velocity versus radial channel position ($y$) for different driving pressures (0.5, 1.0, 1.5, 2.0, 2.5, 3.0 bar). Each data point corresponds to the speed of a single cell. Black lines show individual fit curves obtained by fitting the Cross-model (power-law shear thinning fluid with zero-stress viscosity) to the velocity profile (***Equation 5 - Equation 9***). e, Shear rate of the suspension fluid versus radial channel position ($y$) for different driving pressures. The shear rate is computed with Equation 7. f, Local suspension fluid viscosity at different channel positions computed with ***Equation 6***. g, Suspension fluid viscosity versus shear rate from the fit of the Cross-model (blue line) to the data shown in d, and measured with a cone-plate rheometer (blue circles). h, Tank-treading rotation of a cell in the channel, quantified from the optical flow between two subsequent images. i, Rotational speed of cell image pixels (same cell as in h) versus the ellipse-corrected radius (radial pixel position normalized by the radius of the cell ellipse at that angle). Only cell pixels with an ellipse-corrected radius below 0.7 (dotted line) are used for the linear fit of the tank-treading frequency to the data (solid line) to avoid cell boundary artefacts. j, The angular tank-treading frequency $\omega_{tt}$ increases with the shear rate, with a slope approaching 0.5 for small shear rates (dashed black line). Each point represents the data of an individual cell; different colors indicate different pressures. The red line presents the fit of ***Equation 20*** to the data. k, same as in j but for measurements at a pressure of 2 bar in differently concentrated alginate hydrogels.

The online version of this article includes the following figure supplement(s) for figure 2:

**Figure supplement 1.** Velocity profile of a 2% alginate solution as a function of the $y$-position in the channel for different driving pressures (same data as in ***Figure 2d***).

**Figure supplement 2.** Parameters describing the shear-thinning behavior of the suspension fluid for different alginate concentrations (1.5%, 2.0%, 2.5%).

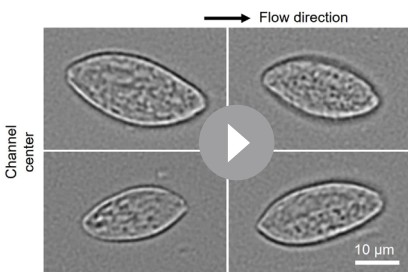

Flow direction

Channel center

10 µm

**Video 1.** Tank-treading motion of cells in a fluid shear flow. Cells are imaged with a frame rate of 500 Hz during their transit through the field-of-view. A smaller image of the cell is cropped from a moving reference frame so that the cell appears stationary. Images are high-pass filtered and contrast-enhanced to visualize cell-internal structures.

https://elifesciences.org/articles/78823/figures#video1

## Tank-treading

The radial velocity gradient of the flow field (the shear rate $\dot{\gamma}$) creates a torque on the sheared and elongated cells and causes them to align in flow direction (*Figures 1e and 2b*) and to rotate in a tank-treading manner (*Video 1*): the cell's elongated shape and alignment angle β remain stationary, but internally, the cell is constantly rotating as if being kneaded between two plates (*Schmid-Schöenbein and Wells, 1969*; *Fischer et al., 1978*).

From a series of images that show the same cells as they flow through the channel, we compute the radial velocity profile $v(y)$ of the fluid flow (*Equation 9*, *Figure 2d*), the shear rate profile $\dot{\gamma}(y)$ (*Equation 7*, *Figure 2e*), and the tank-treading frequency $f_{tt}$ of each cell (*Figure 2h and i*). We find that the tank-treading frequency of a cell is zero at the channel center and increases towards the channel walls (*Figure 2j and k*). At low shear rates (low driving pressure or near the channel center), the rotation rate $\omega_{tt}/\dot{\gamma}$ of individual cells is close to the Einstein-limit of 1/2, as theoretically predicted for spheres that are tank-treading in a Newtonian fluid (*Einstein, 1906*; *Snijkers et al., 2011*; *Roscoe, 1967*). Tank-treading dissipates energy in proportion to the cell's internal viscosity, rotation frequency, and strain. This energy dissipation therefore limits the cell strain in regions of high shear rate and hence shear stress (*Figure 2c*).

## Viscoelastic model

We can quantitatively explain the non-linear strain-stress relationship (*Figure 2c*) and its pressure-dependency by a theoretical framework describing the deformation and alignment of viscoelastic spheres in a viscous fluid under steady shear flow (*Roscoe, 1967*). This theoretical framework (in the following referred to as Roscoe-theory) predicts that the cell strain $\epsilon$ increases proportional with the shear stress σ and the sine of the alignment angle β, and inversely proportional with the elastic modulus $G'$ of the cell (*Equation 16*). The alignment angle β in turn depends on the cell's loss modulus $G''$, the local shear rate $\dot{\gamma}$ and the local shear-dependent viscosity $\eta$ of the suspension fluid (*Equation 17*). With increasing elastic modulus, cells are predicted to deform less (smaller strain $\epsilon$) and to align less in flow direction (larger alignment angle β) when exposed to a fixed shear stress and shear rate. With increasing loss modulus, cells are also predicted to deform less but to align more in flow direction. Thus, from the measurements of cell strain, alignment angle, local shear stress, local shear rate, and local viscosity, Roscoe-theory allows us to compute the viscoelastic properties ($G'(\omega)$ and $G''(\omega)$) of individual cells at twice their specific angular tank-treading frequency, $\omega = 2 \cdot 2\pi f_{tt}$.

## Power-law behavior of cells

When we plot $G'$ and $G''$ of individual cells versus twice their tank-treading frequency $f_{tt}$ (*Figure 3a*), we find that the complex shear modulus $\tilde{G} = G' + iG''$ of a cell population approximately follow a power-law relationship of the form

$$\tilde{G} = k \left( i \frac{\omega}{\omega_0} \right)^{\alpha} \Gamma(1 - \alpha) \tag{1}$$

where $\Gamma$ is the Gamma-function, $k$ is the elastic shear modulus (cell stiffness) referenced to an arbitrarily chosen frequency of 1 Hz by setting $\omega_0 = 2\pi$ rad/s, α is the power-law exponent that characterizes the fluidity of the cell (zero indicating purely Hookean elastic behavior, unity indicating Newtonian viscous behavior), and $i = \sqrt{-1}$ (*Fabry et al., 2001*). Such a behavior of a cell population emerges if the rheology of individual cells also follows a power-law relationship. Thus, using *Equation 1*, we can compare the mechanical behavior of cells measured at different tank treading frequencies by computing their stiffness $k$ (using *Equation 21*) and fluidity α (using *Equation 22*).

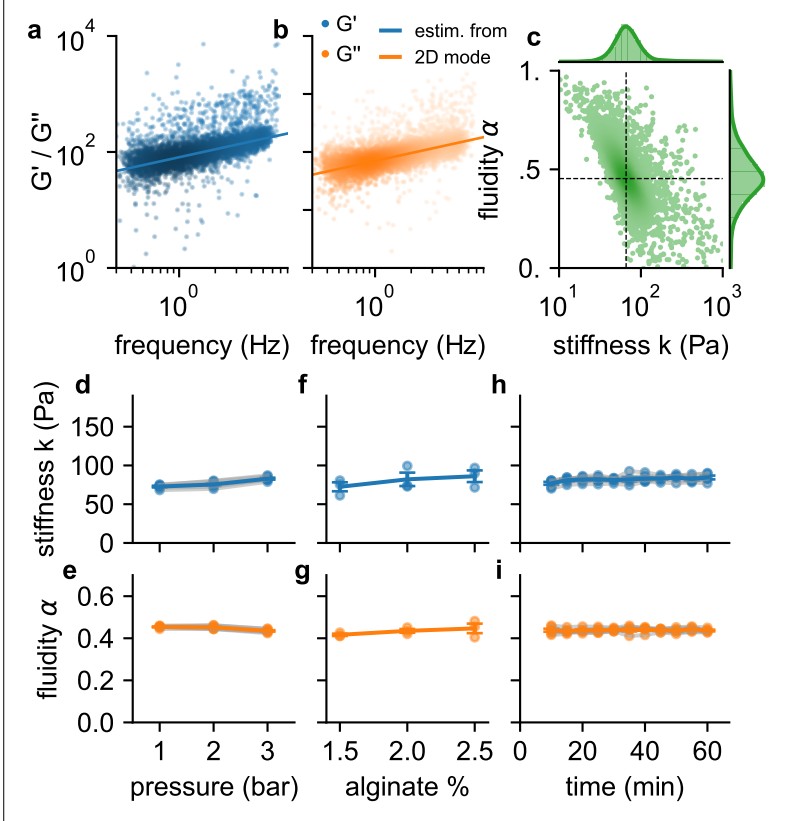

**Figure 3.** Frequency, pressure, suspension fluid, and time dependency of viscoelastic cell behavior. a, $G'$ (blue dots) b and $G''$ (orange dots) of individual NIH-3T3 cells measured at 300 kPa. Lines are not a fit to the data but indicate the predicted behavior of $G'$ (blue line) and $G''$ (orange line) versus angular (tank-treading) frequency according to *Equation 1* of a typical cell with stiffness and fluidity corresponding to the mode of the 2D histogram shown in c (the mode is indicated by the intersection of the dashed lines). c, Distribution of stiffness $k$ and fluidity $\alpha$ of the same cells as shown in a,b, dashed lines indicate the mode of the 2D histogram. Color coding shows 2D Gaussian kernel density estimation. Histograms show the probability density distributions of $k$ (top) and $\alpha$ (side) with Gaussian kernel density estimates (green shading). d, Stiffness $k$ of NIH-3T3 cells increases with pressure (blue lines and symbols indicate mean ± se, gray lines and transparent symbols indicate individual data from 6 independent measurements). e, Fluidity $\alpha$ (same cells as in d) remains constant for all measured pressures. f,g, Stiffness and fluidity show only a weak dependence on alginate concentration (measured at a pressure of 200 kPa, mean ± se (blue) from 3 independent measurements (gray)). h,i, $k$ and $\alpha$ of NIH-3T3 cells remain constant for at least 60 min after suspending them in a 2% alginate solution (measured at a pressure of 300 kPa, mean ± se (blue) from 5 independent measurements (gray)).

The online version of this article includes the following figure supplement(s) for figure 3:

**Figure supplement 1.** Stiffness $k$ and fluidity $\alpha$ of K562 cells (2 independent measurements, measured at a pressure of 3 bar) and THP1 cells (3 measurements, 2 bar) for different cell sizes.

We find in agreement with previous reports (*Desprat et al., 2005*; *Balland et al., 2006*; *Cai et al., 2013*; *Hecht et al., 2015*; *Bonakdar et al., 2016*) that the individual stiffness values $k$ are typically log-normal distributed, and the fluidity values $\alpha$ are normal distributed (*Figure 3b*). Moreover, also in agreement with previous reports, we find an inverse relationship between stiffness and fluidity, whereby stiffer cells tend to be less fluid-like (*Fabry et al., 2001*; *Smith et al., 2005*; *Lange et al., 2015*). Due to this coupling, the mode of the two-dimensional distribution of $\alpha$ and $k$ (the most common combination of $\alpha$ and $k$ among all cells, as estimated from the maximum of the Gaussian kernel-density, *Figure 3b*), provides a robust measure for the mechanical behavior of a cell population.

## Stress stiffening

To test if suspended cells exhibit stress stiffening, as previously reported (*Lange et al., 2017*), we increase the driving pressure from 100 kPa to 300 kPa, which increases the maximum shear stress

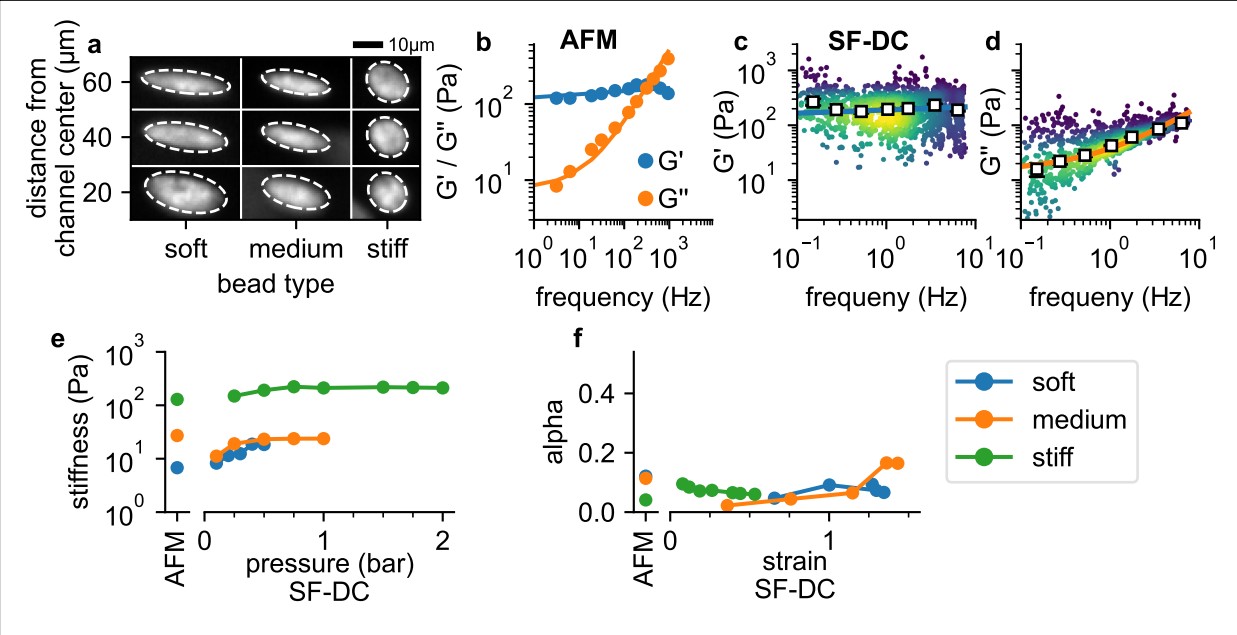

**Figure 4.** Validation with polyacrylamide beads. (**a**) Deformation of PAAm beads with different acralymide-bisacrylamide total monomer concentrations (soft 3.9%, medium 5.9%, stiff 6.9%) at different positions in the channel. (**b**) AFM data (G' and G" versus frequency, mean values from 14 stiff PAAm beads (blue/orange circles), solid lines are the fit of Equation 2 to the data). G' (**c**) and G″ (**d**) for stiff beads at 2 bar. White squares indicate binned median values, blue and orange solid lines are the fit of *Equation 2* to the data. (**e**) AFM-measured stiffness compared to the stiffness versus pressure measured with shear flow deformation cytometry (SF-DC) for differently stiff PAAm beads. (**f**) AFM-measured fluidity compared to fluidity versus strain measured with SF-DC for the same beads as in **e**.

The online version of this article includes the following figure supplement(s) for figure 4:

**Figure supplement 1.** Frequency-dependent shear modulus $G'$ (blue) and loss modulus $G''$ (orange) of polyacrylamide (PAAm) beads (soft: 3.9% $C_{AAmBis}$ (top row); medium: 5.9% $C_{AAmBis}$ (middle row); stiff: 6.9% $C_{AAmBis}$ (bottom row)) measured with atomic force microscopy (left column) and shear flow deformation cytometry (middle and right columns).

at the channel wall from 116 Pa to 349 Pa (*Figure 1f*). Cell fluidity remains constant over this pressure range, but the median stiffness of the cell population increases with increasing pressure by 33% (*Figure 3c and d*). To explore to which extent this stiffness increase is caused by a higher shear stress as opposed to a higher shear rate, we keep the pressure constant at 200 kPa but increase the alginate concentration from 1.5% to 2.5% and therefore the viscosity of the suspension medium from 2.2 Pa·s to 9.2 Pa·s (zero-shear viscosity $\eta_0$ as determined with *Equation 6*). This causes the shear rate to decrease and leads to a slight but not statistically significant increase in stiffness and fluidity (*Figure 3e and f*). Hence, the increase of cell stiffness at a higher driving pressure is induced by stress-stiffening and not by a higher shear rate. We also verify that cell stiffness and fluidity remain stable over a period of up to 60 min after suspending the cells in a 2% alginate solution (*Figure 3g and h*).

## Validation with polyacrylamide beads

To evaluate the accuracy of our method, we measure 16 µm diameter polyacrylamide (PAAm) beads with three different nominal stiffnesses, in a range similar to living cells (*Figure 4a–c*). The frequency-dependency of $G'$ and $G''$ of the beads are calibrated using oscillatory atomic force microscopy (AFM), and conform to a power-law relationship with an additional Newtonian viscosity µ according to

$$\tilde{G} = k \left( i \frac{\omega}{\omega_0} \right)^\alpha \Gamma(1 - \alpha) + i\omega\mu \tag{2}$$

with $\omega_0 = 2\pi$ rad/s (*Figure 4b*). Using shear flow deformation cytometry, we also find a power-law behavior (*Figure 4c and d*). As the maximum frequency remains below 10 Hz in these measurements, however, effect of the Newtonian viscosity term µ is less pronounced (*Figure 4c*), and we therefore perform a global fit of *Equation 2* to the data using a constant µ for all conditions. The values of $k$ and α for beads with different acralymide-bisacrylamid concentrations are comparable between AFM and

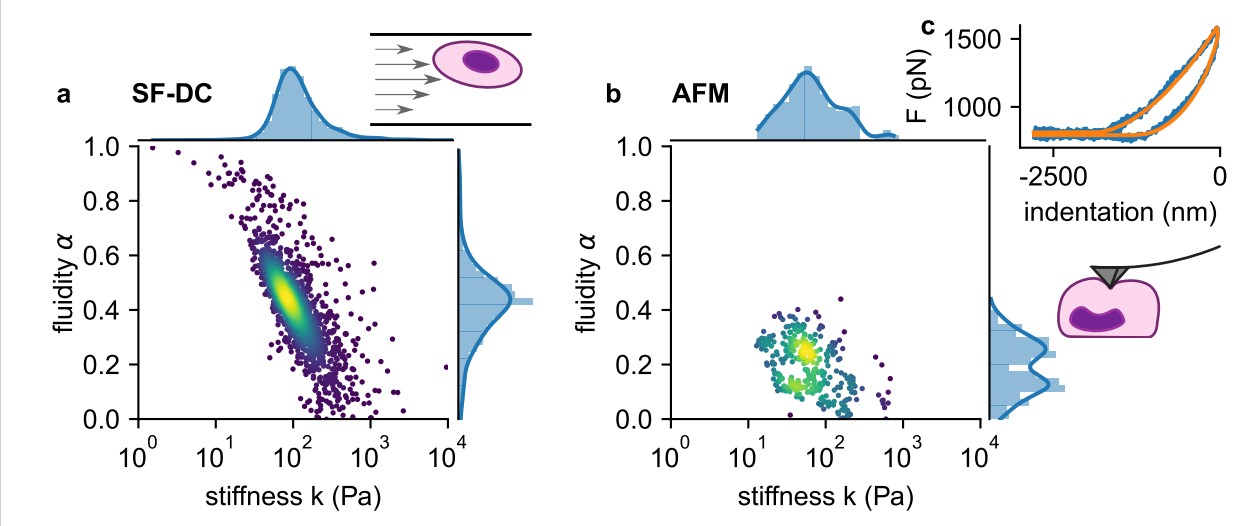

**Figure 5.** Comparison of viscoelastic cell properties measured with shear flow deformability cytometry (SF-DC) and AFM. a. Stiffness $k$ versus fluidity $\alpha$ of THP1 cells (n=5000) measured with SF-DC at a pressure of 2 bar, 2% alginate solution. Colors represent Gaussian kernel density. Histograms show the probability density distributions of $k$ (top) and $\alpha$ (side) with Gaussian kernel density estimates (blue line). b, AFM measurements of THP1 cells. Each point represents $k$ and $\alpha$ from one cell, each obtained from the fit of **Equation 24** to 3 or more force-indentation curves for each cell. c, Typical force-indentation curve (blue line) and fit with **Equation 24** (orange line).

shear flow deformation cytometry measurements (**Figure 4—figure supplement 1**). Moreover, $k$ and $\alpha$ are largely pressure-independent (from 0.2 to 2 bar; **Figure 4e**), as expected for a linear material such as PAAm. Fluidity is close to zero for strains below unity ($\alpha$=0.092 for 5.9% $C_{AAmBis}$, and $\alpha$=0.074 for 3.9% $C_{AAmBis}$), indicating predominantly elastic behavior as expected. Fluidity increases slightly at higher strains (**Figure 4f**), likely due to fluid-induced (poroelastic) relaxation processes (**Kalcioglu et al., 2012**). Together, these results demonstrate that our method provides quantitatively accurate estimates for the elastic and dissipative properties of soft spherical particles.

We next compare the viscoelastic properties of monocytic THP-1 cells probed by shear flow cytometry and atomic force microscopy (AFM). We acquire force-indentation curves at rates of ~1 /s (**Figure 5c**), which is within the range of strain rates that cells experience in our shear flow cytometry setup. AFM measurements show that THP-1 cells conform to power-law rheology with an additional Newtonian viscosity term according to **Equation 2**, from which we extract the shear modulus $k$ and fluidity $\alpha$ (**Figure 5b**). THP-1 cells appear stiffer (at 1 Hz) and more fluid-like when measured with shear

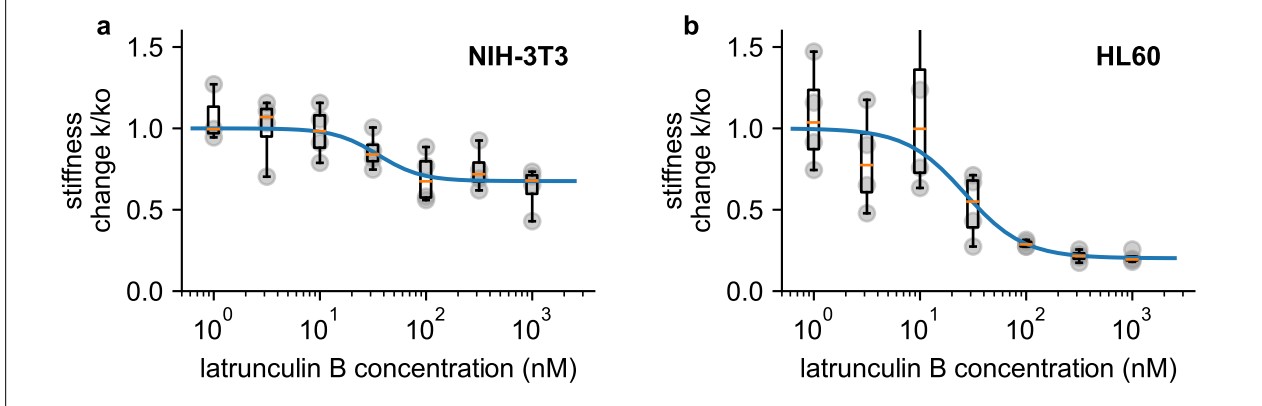

**Figure 6.** Dose response measurements. a, Dose response curve of NIH-3T3 cells treated with different concentrations of latrunculin B. Stiffness is normalized to the stiffness of DMSO-treated cells, grey points indicate n=4 independent measurements for each concentration, each measurement is the average of a 0.5, 1, 2, and 3 bar measurement, boxplot indicate median (orange line) and 25 and 75 percentiles, whiskers indicate 5 and 95 percentiles. Blue line is the fit of the Hill-Langmuir-equation to the data, with an EC50 of 31.2 nM. b, Dose-response curve of HL 60 cells. Hill-Langmuir fit gives an EC50 value of 25.9 nM.

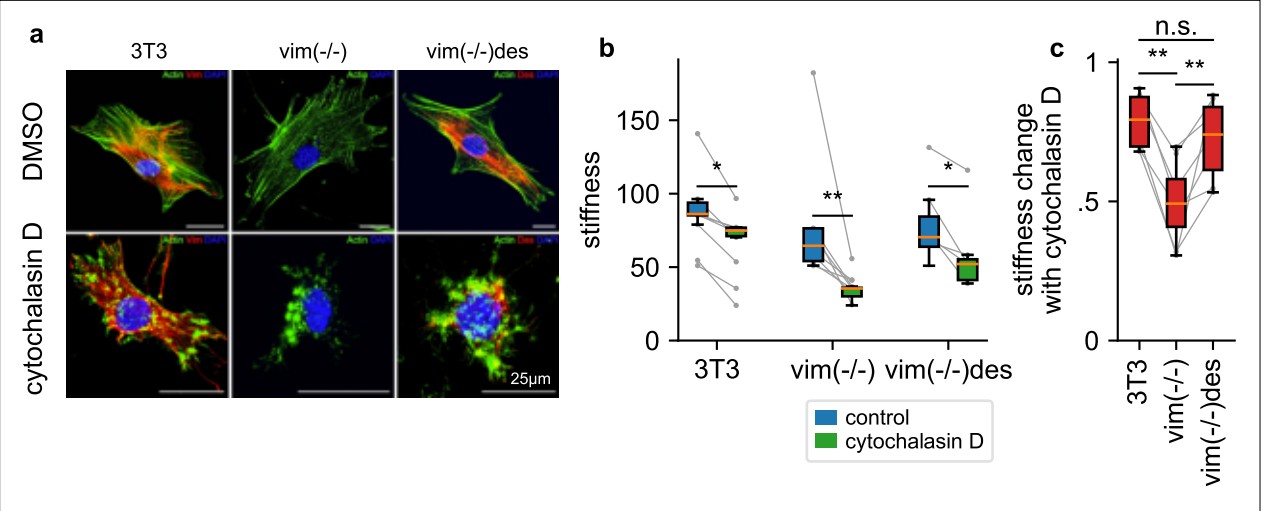

**Figure 7.** Influence of intermediate filaments. a, NIH-3T3 MEFs, vimentin-knockout, and desmin-knockin MEFs in DMSO control conditions (upper row) and with cytochalasin D treatment (lower row). Actin (stained with phalloidin-Atto-488) is shown in green, desmin (stained with a rabbit-anti-desmin-CT1 antibody) in red, and the nucleus (stained with DAPI) in blue. b, Stiffness of DMSO control (blue box) and cytochalasin D treated cells (green box) (orange line: median, box: 25 and 75 percentile, whiskers: 5 and 95 percentile, gray points and lines connect mean values from independent measurements performed on the same day, with 1813 cells on average contributing to each data point). c, Stiffness change after treatment with cytochalasin D relative to DMSO control. (statistical significance: * p < 0.05, ** p < 0.01, two-sided Mann-Whitney-U test).

flow cytometry ($k$=82 Pa, $\alpha$=0.44) compared to AFM ($k$=52 Pa, $\alpha$=0.25). Despite these differences, AFM measurements confirm the applicability of power-law rheology, and they also show a log-normal distribution of cell stiffness $k$ as well as an inverse relationship between $k$ and fluidity α as seen in our shear flow measurements (***Figure 5a and b***).

## Dose-response measurements

We perform dose-response measurements using latrunculin B (LatB), which prevents the polymerization of monomeric actin and leads to a depolymerization of the actin cytoskeleton (***Urbanska et al., 2020***). NIH-3T3 fibroblasts soften with increasing doses of LatB (1–1000 nM) according to a sigmoidal (Hill-Langmuir) relationship, with a maximum response of 1.47-fold and a half-maximum dose of EC50=35.2 nM (***Figure 6a***). These responses agree with published data obtained using real-time deformability cytometry (RT-DC) measurements on HL-60 cells (maximum response 1.46-fold, EC50=26.5 nM) (***Urbanska et al., 2020***). When we measure pro-myoblast HL-60 suspension cells with our setup, EC50 is similar to published data (26.4 nM), but the maximum response is much higher (5.0-fold) (***Figure 6b***).

## Role of intermediate filaments

To explore the attenuated LatB responsiveness of NIH-3T3 fibroblasts compared to HL-60 leukemia cells, we reasoned that NIH-3T3 cells express high levels of the intermediate filament protein vimentin (***Figure 7a***) that may protect the cells from excessive deformations when filamentous actin is depolymerized. To test this idea, we measure the stiffness of NIH-3T3 and vimentin-knock-out (vim(-/-)) fibroblasts in response to 30 min treatment with cytochalasin D (2 µM), which binds to the barbed end of filamentous actin and—similar to LatB—leads to a net depolymerization of the actin cytoskeleton (***Figure 7a***). The NIH-3T3 cell line has been established from mouse embryonic fibroblasts (MEFs) by spontaneous immortalization (***Todaro and Green, 1963***). We followed the corresponding protocol for MEFs obtained from vimentin-knockout mouse embryos (***Colucci-Guyon et al., 1994***). Thus, the three cell lines investigated here are of the same cell type. We find that cytochalasin D treated vim(-/-) cells soften by a considerably greater extent (2.16-fold) compared to wild-type cells (1.22 fold) (***Figure 7b and c***), in support of the notion that vimentin stabilizes the cytoskeleton.

To explore if the cytoskeleton-stabilizing effect of vimentin is a general feature also of other intermediate filament networks, we measure the cytochalasin D response of desmin-transfected vimentin

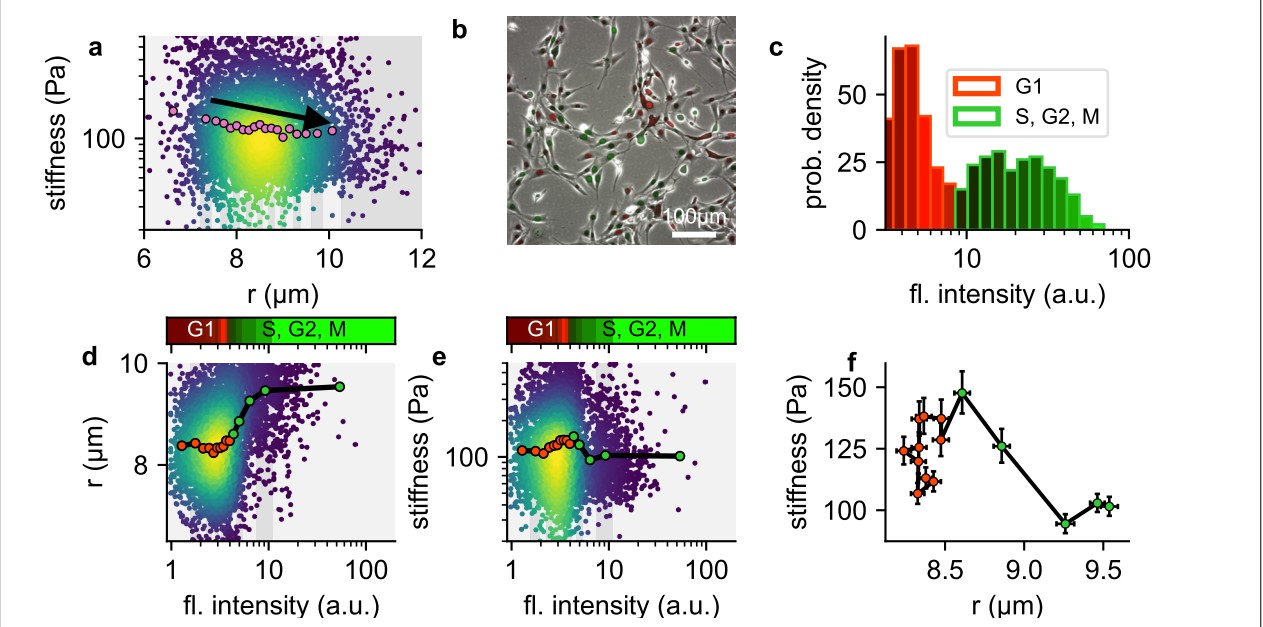

**Figure 8.** Influence of cell cycle. a, Cell stiffness versus cell radius, each point corresponds to data from one cell, colors represent Gaussian kernel density, pink circles show median values over bins with equal cell count. Cell stiffness tends to decrease with increasing cell radius. b, Phase contrast and fluorescent image of Fucci-cell cycle indicator-transfected NIH-3T3 cells. Cells in G1 phase show low green and high red fluorescence intensities, cells in S, G2, or M phase show high green and low red fluorescence intensities. c, Histogram of green fluorescence intensities of Fucci-cell transfected NIH-3T3 cells. Bar colors reflect the RGB-colormap of the red and green channel intensities averaged over all cells within a bin. Accordingly, the cell cycle can be deduced from the green intensity alone. d, Cell radius versus green fluorescent intensity. Each point corresponds to data from one cell, colors represent Gaussian kernel density, circles show median values over bins containing an equal number (~100) of cells. Colorbar represents the RGB-colormap of the red and green intensities of the cells before harvesting, mapped onto the green fluorescent intensity after harvesting measured in the shear flow cytometer. Cell radius increases after cells exit G1 phase. e, Cell stiffness versus green fluorescent intensity. Cells stiffness increases during G1 phase and decreases after entering S phase. f, Cell stiffness versus cell radius; data points correspond the the median values in d and e, red color designates cells in G1 phase, green color designates cells in S, G2 or early M phase. During G phase, cells increase their stiffness while maintaining their radius. After entering S phase, cells increase their radius while their stiffness decreases.

knock-out MEFs (vim(-/-)des). Desmin, which is the dominant intermediate filament in skeletal muscle, forms an intermediate filament network in fibroblasts that is structurally similar to the vimentin network in wild-type cells (*Figure 7a*). Similar to vimentin-expressing MEFs, vim(-/-) desmin-expressing MEFs also display an attenuated cytochalasin D response (1.37-fold), confirming that both the vimentin and desmin intermediate filament network can protect cells from excessive deformations when filamentous actin is depolymerized (*Figure 7c*).

## Cell cycle dependence

In our measurements, we observe that larger NIH-3T3 cells tend to be softer compared to smaller cells (*Figure 8a*). We hypothesized that this weak size-dependence of cell stiffness might be attributable to cell cycle progression, which leads to changes in chromatin compaction and cell volume. To test this hypothesis, we extend our setup to acquire green fluorescent images alongside bright field images of cells transfected with a two-color fluorescent Fucci cell cycle indicator (*Sakaue-Sawano et al., 2008*). Fucci-transfected cells display high red and low green fluorescence when they are in G1 phase, and low red but increasing levels of green fluorescence as they progress into S, G2, and early M-phase (*Sakaue-Sawano et al., 2008*). We measure the cell cycle distribution of NIH-T3T cells before harvesting using epifluorescence microscopy (*Figure 8b*), and map the distribution to the green fluorescent intensities measured in our shear flow cytometry setup (*Figure 8c*).

We find as expected that cell radius increases with cell cycle progression (*Figure 8d*). In addition, cell stiffness steadily increases towards the end of the G1 and the beginning of the S-phase, and then rapidly decreases as the cell cycle progresses (*Figure 8e*). When we bin the cells according to their green fluorescent intensities (i.e. according to their cell cycle progression) and plot stiffness versus cell

radius (*Figure 8f*), we find substantially larger and non-monotonic fluctuations of cell stiffness versus cell radius, compared to the smaller, monotonic decrease of cell stiffness in the radius-binned data (*Figure 8c*). These differences arise because changes in cell stiffness and cell radius occur at different stages of the cell cycle.

## Discussion

Viscoelastic cell properties can be measured with established methods such as atomic force microscopy (*Cordes et al., 2020*), micropipette aspirations (*Zhelev et al., 1994*), or magnetic tweezer microrheology (*Bonakdar et al., 2016*). These methods have a relatively low throughput of typically below 10–100 cells/hr. The need to measure cell mechanical properties with substantially higher throughput led to the recent development of various microfluidic techniques (*Urbanska et al., 2020*) including hydrodynamic stretching (*Gossett et al., 2012*), real-time deformability cytometry (*Otto et al., 2015*; *Fregin et al., 2019*), micro-filtration (*Rowat et al., 2012*), and micro-constriction systems (*Lange et al., 2015*; *Lange et al., 2017*).

Our method builds on previously established high-throughput microfluidic approaches, with several modifications: We suspend cells in a medium that is pumped with high pressure (typically 50–300 kPa) through a long, parallel microfluidic channel with one inlet and outlet (no flow-focussing geometry is needed). Such simple microfluidic channels are commercially available at low cost, which we expect will facilitate a widespread adoption of the technique. The large driving pressure gives rise to sufficiently large (>50 Pa) shear stresses to induce measurable cell deformations. The high pressure can be controlled with a simple pressure regulator, without the need for a precise microfluidic controller—another advantage compared to existing methods that typically operate under lower pressure. The width and height of the channel (200 µm) are much larger than the cell diameter, which prevents clogging due to debris that is often encountered in microfluidic constriction-based systems (*Lange et al., 2015*; *Lange et al., 2017*). Most importantly, the large channel diameter ensures that fluid shear stresses do not vary appreciably across the cell, which simplifies the analysis of cell mechanical properties as the cells do not deform into complex bullet- or hourglass-like shapes as seen in other methods (*Lange et al., 2015*; *Lange et al., 2017*; *Urbanska et al., 2020*). By suspending the cells in a fluid with high viscosity (typically >1 Pa·s), we achieve a flow speed that is sufficiently low (<20 mm/s) so that the cells' speed, position, and shape can be captured without motion blur at a typical exposure time of 30 µs using a standard CMOS-camera mounted to a routine laboratory microscope.

The lateral flow profile in the channel causes a tank-treading-like cell motion, which imposes periodic cell deformations with frequencies on the order of 10 Hz. At such low frequencies and strain rates, elastic cell properties dominate over viscous cell properties (*Fabry et al., 2001*; *Mietke et al., 2015*). The cell transit through the microfluidic channel lasts for several seconds, which is much longer than the period time of the cells' tank treading rotation, implying that the measured cell deformations can safely be assumed to have reached a steady-state. Measuring visco-elastic parameters from steady-state cell deformations has a major advantage over existing microfluidic techniques in that no visco-elastic models (e.g. Maxwell, Voigt etc.) or pre-conceived response functions (e.g. exponential, bi-exponential, power-law etc.) are needed to fit the transient cell deformation responses during the cells' passage through narrow constrictions and channels (*Lange et al., 2015*; *Fregin et al., 2019*). If for example an exponential function is fitted to a power-law creep response, the resulting viscoelastic cell properties would largely depend on the time scale of the experiment (e.g. the passage time of the cell through the microfluidic constriction or channel) and less so on the cell's intrinsic mechanical properties (*Fabry et al., 2001*; *Lange et al., 2015*; *Fregin et al., 2019*) By contrast, the values obtained with our method are not influenced by the time scale of the experiment.

From images of the same cell as it is flowing through the channel, we estimate the tank treading frequency and the flow velocity; from the flow velocity profile across the channel, we compute the local shear rate (*Equation 5*) and the local shear-dependent viscosity of the suspension fluid (*Equation 6*); from the radial cell position, we compute the local shear stress (*Equation 4*); from the cell shape, we compute the strain (*Equation 10*) and the alignment angle in flow direction. From these measurements, we finally compute the cell's viscoelastic properties (stiffness and fluidity, *Equation 21 and 22*). Hence, once the flow velocity profile is known, we can determine the viscoelastic properties from a single image because all cell deformations are in a steady-state.

We provide user-friendly software for image acquisition and data analysis on a standard PC, which can be downloaded at https://github.com/fabrylab/shear_flow_deformation_cytometer (copy archived at swh:1:rev:2d35a697243c432cddd52e10d2e3c5121f922adf; *Gerum, 2022*). Currently, the method stores the acquired uncompressed images on a hard drive, which in the case of typically 10,000 images for a single experiment lasting 20 s amounts to a storage space of nearly 4 GB. The image data are analyzed afterwards, which at a rate of around 50 images per second can take several minutes. Future software developments and faster computer hardware will enable image analysis on the fly for real-time shear flow deformation cytometry.

The computation of viscoelastic properties is based on a theoretical model proposed by R. Roscoe that describes the deformation of homogeneous, isotropic, incompressible neo-Hookean viscoelastic spherical particles under fluid shear stress (*Roscoe, 1967*). Cells in suspensions, however, are known to deform non-linearly (*Lange et al., 2017*), with stress- or strain stiffening that is more pronounced than the stiffening predicted for a neo-Hookean material. Therefore, our measurements represent an effective secant modulus and not a small-strain tangential modulus. Moreover, cells do not consist of a homogeneous material but of different components (e.g. the cell cortex and the nucleus) with different mechanical properties (*Zhelev et al., 1994*; *Rowat et al., 2012*; *Mietke et al., 2015*; *Cordes et al., 2020*). As a consequence, cells do not always deform into ellipsoidal shapes but occasionally deform into sigmoidal shapes, which becomes more pronounced in response to larger shear stresses or drugs that soften the cytoskeleton, such as cytochalasin D or latrunculin B.

Despite the simplified assumptions of the Roscoe theory, however, our cell rheological measurements agree with previously published findings that were obtained using a range of different methods and models, namely that suspended cells show a behavior that is consistent with power-law rheology, that the elasticity of individual cells is log-normal distributed, that the fluidity of individual cells is normal-distributed, and that stiffness and fluidity scale inversely (*Fabry et al., 2001*; *Alcaraz et al., 2003*; *Desprat et al., 2005*; *Lange et al., 2015*). These experimental findings are in agreement with predictions from soft glassy rheology (*Sollich, 1998*; *Fabry et al., 2001*). Moreover, we show that stiffness and fluidity values of polyacrylamide beads and cells measured with shear flow deformation cytometry agree quantitatively with AFM measurements.

Our measurements are insensitive to changes in the viscosity of the suspension medium, demonstrating that the fluid-mechanical assumptions of the Roscoe theory hold in the case of living cells in a shear-thinning suspension fluid. We find that cells appear stiffer when measured at higher driving pressures, likely due to stress- or strain-stiffening of the cells (*Lange et al., 2017*). When we measure linearly elastic polyacrylamide beads over a 10-fold pressure range (from 20 to 200 kPa), we see a constant, pressure-independent shear modulus and agreement with the stiffness and fluidity values measured using AFM, demonstrating that the Roscoe theory gives quantitatively accurate estimates, regardless of driving pressure and suspension fluid viscosity.

Roscoe theory estimates the cell viscosity relative to the viscosity of the suspension fluid, which for a shear thinning fluid such as alginate can be difficult to measure. However, since we know the fluid profile in the microfluidic channel (from the flow speed of hundreds of cells), we can estimate the rheological properties of the suspension fluid, including its shear thinning behavior. This ability is more than a by-product of our method and could be valuable for example for researchers interested in bioink development for applications in biofabrication. Moreover, we measure the complex rheology of the suspension fluid at the very same time and under the same conditions (temperature, range of shear rates) as the cells. Therefore, unlike other microfluidic cytometry methods (*Otto et al., 2015*; *Fregin et al., 2019*), our quantification of visco-elastic cell properties does not rely on separate measurements using cone-plate or other types of rheometers. The rheological parameters of alginate solutions measured with our method closely agree with cone-plate rheometer measurements, with relative deviations of 31% over a shear rate spanning 5 orders in magnitude (from 0.01 to 1000 $s^{-1}$).

Our method measures each cell at a single tank-treading frequency that depends on the cell's lateral position in the channel. Thus, with our method we sample the frequency-dependent mechanical properties of a cell population simply by observing cells at different channel positions. By contrast, with existing methods, time- or frequency-dependent cell responses can only be explored by choosing different strain rates, for example via adjusting the driving pressure (*Lange et al., 2015*). The tank-treading frequency can be directly measured using particle flow analysis methods in a subset of the cells that shows small features with high contrast (*Fischer et al., 1978*). For the remaining cells, it is

possible to estimate the tank treading frequency from the local shear rate according to an empirical equation (*Equation 20*). This equation holds for the cell types and suspension fluids used in our study, but we do not claim that it holds universally for other cell types or suspension fluids. For example, red blood cells exhibit a tumbling instead of a tank-treading motion at low shear rates (*Schmid-Schöenbein and Wells, 1969*), and *Equation 20* underestimates their tank-treading frequency at shear rates beyond $100\,s^{-1}$(*Fischer et al., 1978*).

Shear flow deformation cytometry
for measuring visco-elastic cell properties

1 Cell suspension m        Cell preparation
2 Device set-up
3 Running the experiment
4 Analyzing the data

**Video 2.** Protocols and instructions for shear flow deformation cytometry. The video explains step-by-step how to prepare cells for measurements, how to set up the measurement device, and how to operate the data acquisition software.
https://elifesciences.org/articles/78823/figures#video2

To demonstrate its practical applicability, we apply our method to measure the stiffness of HL-60 cells in response to different doses of the actin-depolymerizing agent latrunculin B. We find in agreement with previous observations a half-maximum dose (EC50) of around 30 nM, but a considerably larger softening of the cells by a factor of 5.4-fold at the highest dose of 1 μM, compared to a softening of only 1.5 fold that is seen with other microfluidic techniques constriction microfluidic constriction-based deformability cytometry (cDC), and real-time deformability cytometry (RT-DC) (*Urbanska et al., 2020*). This higher responsiveness is likely attributable to the relatively low cellular strain rates in our method, which are on the order of $10\,s^{-1}$, compared to strain rates of around $100\,s^{-1}$ in the case of RT-DC. At these high strain rates, viscous cell behavior starts to dominate over cytoskeleton-associated elastic behavior (*Fabry et al., 2001*; *Mietke et al., 2015*). Accordingly, when cells are measured with extensional flow deformability, a method that operates at even higher strain rates in the kHz-range, they do not appreciably soften in response to LatB (*Urbanska et al., 2020*; *Gossett et al., 2012*).

We also demonstrate that the cell softening induced by cytochalasin D, another actin-depolymerizing drug, is attenuated in the presence of intermediate filaments (vimentin or desmin), and becomes more pronounced when intermediate filaments are absent. This finding is in line with earlier reports that intermediate filaments protect cells against excessive strain (*Patteson et al., 2020*), and that the absence of vimentin in fibroblasts leaves the cells vulnerable to mechanical stress (*Eckes et al., 1998*). A physical interaction between vimentin intermediate filaments and F-actin bundles as mediated by plectin has been demonstrated by high resolution immuno-electron microscopic methods (*Svitkina et al., 1996*). The importance of vimentin-actin interactions has furthermore been corroborated by high resolution structured illumination microscopy in combination with cryo-electron tomography, revealing the intimate association and functional crosstalk between vimentin intermediate filaments and F-actin stress fibers (*Wu et al., 2022*). Here, we have directly demonstrated that the stable introduction of a cytoplasmic intermediate filament protein into intermediate filament-free cells restores their cytoskeletal functionality and mechanical stability.

Shear stress deformability cytometry can be combined with fluorescent imaging. Here, we image the viscoelastic properties of NIH-3T3 cells together with the cell cycle using the fluorescent Fucci indicator. Our data demonstrate that NH-3T3 cells stiffen during the course of cell cycle progression in G1 phase, with a maximum stiffness during late G1 – early S-phase, and then soften before they enter the G2 and M-Phase. Since cell volume also increases during the transition from G1 to S phase, we find a slight overall dependence of cell stiffness on cell size in the case of NIH-3T3 cells (*Figure 8c*). This cell size dependence is also detectable in HL-60 and THP1-cells (*Figure 3—figure supplement 1*).

In summary, shear flow deformation cytometry provides accurate quantitative measurements of elastic and dissipative cell properties at high throughput. The method can be easily and inexpensively implemented on standard or research grade microscopes. Unlike other high-throughput microfluidic methods, the cells are measured under near steady-state conditions at low to moderate strain rates where elastic responses dominate over viscous responses.

**Table 1.** Cell line-specific composition of culture medium.

| cells | Base medium | serum | PenStrep | GlutaMAX | Geneticin | Sodium pyruvate | MEM NEAA | HEPES |
|---|---|---|---|---|---|---|---|---|
| NIH-3T3 | DMEM | 10% BCS | 1% | - | - | - | - | - |
| vim(-/-) | DMEM | 10% FCS | 1% | 1% | - | - | - | - |
| vim(-/-)des | DMEM | 10% FCS | 1% | 1% | 1 mg/ml | - | - | - |
| HL60 | RPMI | 10% FCS | 1% | - | - | - | - | - |
| THP-1 | RPMI | 10% FCS | 1% | - | - | 1mM | 1% | 10mM |

All % values are (v/v)%. NIH-3T3: mouse embryonic fibroblast cells (No. CRL-1658, American Type Culture Collection); vim(-/-): mouse embryonic fibroblast cells derived from vimentin(-/-) mice (kindly provided by Prof. Dr. T. M. Magin, University of Bonn) and further subcloned to eliminate desmin- or keratin-expressing cells, as described in *Gregor et al., 2014*; vim(-/-)des: vim(-/-) MEFs re-expressing wild-type desmin, generated as described in *Herrmann et al., 2020*; HL60: human leukemic lymphoblast cells (No. CCL-240, American Type Culture Collection); THP-1: monocytic cells (No. TIB-202, American Type Culture Collection); DMEM: Dulbecco's Modified Eagle Medium (Gibco 11995065); RPMI: Roswell Park Memorial Institute 1640 Medium (Gibco 21875034); PenStrep: 100 x penicillin-streptomycin-glutamin solution (Gibco 10378016); GlutaMAX: L-alanine-L-glutamine supplement (Gibco 35050–038); Geneticin (Gibco 10131027); BCS: bovine calf serum (Sigma 12,133 C); FCS: fetal calf serum (Sigma F7524); MEM NEAA: 100 x non-essential amino acid solution without L-glutamine (Gibco 11140–035); HEPES (Gibco 15630–056); Sodium pyruvate (Gibco 11360–039).

## Methods

The measurement setup is depicted in *Figure 1* a. *Video 2* explains the measurement procedure. Cells are suspended in a high-viscosity medium (e.g. a 2% alginate solution), and are pressed via a 10 cm long, 1mm inner diameter silicone tube through a 5.8 cm long microfluidic channel with a square cross section of 200x200 µm (CS-10000090; Darwin Microfluidics, Paris, France). The driving air pressure of typically 1–3 bar is regulated with a pressure regulator (KPRG-114/10, Knocks Fluid-Technik, Selm, Germany) and can be switched on or off with a three-way valve (VHK2-04F-04F; SMC, Egelsbach, Germany). The air pressure is measured with a digital pressure gauge (Digi-04 0.4%, Empeo, Germany). Cells flowing through the channel are imaged in bright-field mode at 50–500 Hz (depending on the flow speed) with a CMOS camera (acA720-520um, Basler, Germany) using a 40x0.4 NA objective (Leica) in combination with a 0.5 x video coupler attached to an inverted microscope. After passing the microchannel, the cells are collected in a waste reservoir.

### Cell culture

Cells are cultured at 37 °C, 5% $CO_2$ and 95% humidity and are split every 2–3 days for up to 20 passages.

### Preparing cells for rheological measurements

Our method for measuring viscoelastic cell properties requires that the cells, if they are adherent to a cell culture dish (NIH-3T3, vim(-/-), vim(-/-)des), are brought into suspension. For cells grown in 75 cm² flasks, we remove the medium and wash the cells three times with 10 ml of 37 °C PBS. After removing the PBS, 5 ml of 0.05% trypsin/EDTA in PBS are added and distributed over the cells, and after 10 s, 4 ml of the supernatant are removed. Cells are then incubated for 3–5 min at 37 °C, 5% $CO_2$. 5 ml of 37 °C cell culture medium (*Table 1*) are added to the flask, and the cells are counted. If cells are already in suspension (THP-1 and HL60 cells), the above steps are omitted. $10^6$ cells are taken out of the flask, centrifuged for 5 min at 25 rcf (NIH-3T3, vim(-/-) and vim(-/-)des) or 290 rcf (HL-60 and THP-1) to remove the supernatant, gently mixed in 1 ml of equilibrated suspension fluid (see below), transferred to a 2 ml screw-cup test tube, and centrifuged at 150 rcf for 30 s to remove air bubbles.

### Suspension fluid preparation

Alginate solution is prepared freshly for the next day. Sodium alginate powder (Vivapharm alginate PH176, batch nr. 4503283839, JRS Pharma GmbH, Rosenberg, Germany, or alginic acid sodium salt from brown algae, A0682, Sigma Aldrich, for THP1 cells) is dispersed at a concentration of 1.5%, 2%, or 2.5% (w/v) in serum-free cell culture medium (*Table 1*). The alginate solution is mixed overnight with a magnetic stirrer at room temperature until all powder has been dissolved. The suspension fluid

is then equilibrated by incubating for 6 hr at 37 °C, 5% $CO_2$. When prepared with RPMI media (but not when prepared with DMEM nor Sigma Aldrich alginate), the alginate solution is filtered with a 0.45 µm filter before use. 1 ml of alginate solution are then added to the cell pellet of $10^6$ cells in the Falcon tube and mixed using a positive displacement pipette (15314274, Gilson/Fisher Scientific) by slowly (~2 s cycle time) and repeatedly (10 x) sucking the liquid in and out. The alginate-cell suspension is then transferred into a 2 ml screw-cup test tube and centrifuged for 30 s at 150 rcf to remove air bubbles.

## Drug treatment

Drugs are mixed in the alginate for at least 15 min with a magnetic stirrer at 350 rpm inside an incubator (37 °C, 5% $CO_2$, 95% relative humidity) prior to mixing-in the cells. Cells are prepared as described above and mixed with the alginate-drug mixture using a positive displacement pipette by slowly (~2 s cycle time) and repeatedly (10 x) sucking the liquid in and out. The alginate-drug-cell suspension is transferred into a 2 ml screw-cup test tube and incubated for a prescribed time at 37 °C, 95% rH. Prior to measurements, the alginate-drug-cell suspension is centrifuged at 150 rcf for 30 s to remove air bubbles.

Inhibition of actin polymerization on NIH-3T3, vimentin-knockout and desmin-knockin MEFs is performed with cytochalasin D (Cat. No. C8273; Sigma-Aldrich, St. Louis, MO). Cytochalasin D is dissolved in DMSO at a stock concentration of 20 mM. The equilibrated alginate (3 ml) is either mixed with cytochalasin D to a final concentration of 2 µM, or mixed with DMSO to a final concentration of 0.01% (DMSO control), or mixed with 3 µl of DMEM (negative control). Cells harvested from a single cell culture flask are split into three groups of $10^6$ cells, each group is suspended in one of the alginate solutions as described above, stored in an incubator for 15 min (alternating between either negative control of DMSO control), 30 min (drug-treated), and 45 min (alternating between either DMSO control or negative control), and measured.

Inhibition of actin polymerization on NIH-3T3 cells is performed with latrunculin B (LatB, Cat. No. L5288; Sigma-Aldrich, St. Louis, MO, dissolved in DMSO at a stock concentration of 2 mM). We add 2 µl of LatB (stock) or 2 µl of DMSO to 4 ml of alginate (final concentration 1000 nM LatB, 0.2% DMSO), and mix with a magnetic stirrer at 350 rpm for 15 minutes. 1850 µl of the alginate-drug mixture is then added to 4 ml of alginate, mixed for 15 min, and the process is repeated to obtain a dilution series with LatB concentrations of 1000, 316, 100, 32, 20, 3.2, and 1 nM. The alginate-DMSO mixture is diluted in the same way. Cells are prepared and mixed into the alginate as described above and stored at room temperature for 10 min (LatB) or 20 min (DMSO control) prior to measurements.

## Image acquisition

Typically, 10,000 images per measurement are recorded with a CMOS camera (acA720-520um, Basler, Germany) at a frame rate of 50–500Hz with an exposure time of 30 µs. To measure the flow speed, each cell has to be recorded in at least 2 consecutive images. Therefore, the frame rate $fr$ is chosen depending on the maximum flow speed $v_{max}$ and the width of the region of interest (ROIx): $fr > v_{max} / (0.5 \text{ ROIx})$. In our setup, the ROIx is 248µm, resulting in a maximum flow speed of 41mm/s for a frame rate of 500Hz. To prevent motion blur, however, we keep the maximum flow speed to about 20mm/s.

Fluorescent images can be acquired in parallel with the bright field images. A 300 mW diode-pumped solid-state laser (wavelength 473nm, VA-I-N-473; Viasho, Beijing, China) serves as an epifluorescent light source, and a beam splitter projects the bright field and fluorescent images onto two synchronized cameras. To separate the light paths, the bright-field illumination is long-pass filtered (>590nm), and a band-pass filter (500–550nm) is placed in front of the camera for the fluorescent channel.

We provide software for image acquisition (see below under Software flow chart), which includes a live-viewer and user-friendly interface for entering meta information (e.g. applied pressure, suspension medium, drug treatments) and configuration settings (e.g. frame rate, total number of images to be stored). The software is based on the pypylon library to record the images, and Python (***Van Rossum and Drake, 2009***) and Qt to provide the user interface.

## Cell shape analysis

We normalize the bright-field images by subtracting the mean and dividing by the standard deviation of the pixel intensities. A neural network (U-Net *Ronneberger et al., 2015*, tensorflow *Abadi, 2016*) trained on labeled images of different cell types and suspension media detects the cell outline and generates a binary mask, to which an ellipse is fitted ($x,y$ position of the ellipse center, its semi-major ($a$) and semi-minor axis ($b$), and the angle of orientation β of the major axis with respect to the flow ($x$) direction, see *Figure 1d and e*, *van der Walt et al., 2014*). Binary masks that do not conform to an elliptical shape based on circumference or solidity criteria (e.g. due to cell doublets or erroneous cell outlines due to poor image contrast) are discarded.

## Finding the channel mid plane and center line

Prior to recording the images, the microscope must be precisely focused to the mid plane (z=0, see *Figure 1b*) of the channel. To do so, we apply a small pressure (50–100 Pa) to the suspended cells and focus the microscope in phase contrast mode to the bottom of the microchannel, which can be unambiguously identified by stationary or very slowly flowing small debris. We then move the objective up by 75 μm, which corresponds to half the microchannel's height (100 μm) divided by the refractive index of the suspension medium. We confirmed that the reproducibility of the method is within ±1.7 μm (rms) when a 40x0.6 NA objective is used.

The channel center line ($y = 0$, see *Figure 1b*) is identified from the flow speed profile as a function of the radial ($y$) position. Flow speed is computed by tracking cells over subsequent images and dividing the distance they have moved in $x$-direction by the time difference between images. A polynomial of the form.

$$v(y) = v_{\max} \left( 1 - \left| \frac{y - y_c}{W/2} \right|^{\zeta} \right) \tag{3}$$

is then fitted to the velocity profile to identify the center position of the channel ($y_c$), with the maximum flow speed $v_{max}$ at the channel center as the second fit parameter, and the exponent $\zeta$ as the third fit parameter. $W$ is the channel width. The fit parameter $y_c$ is then used to shift the image y-coordinate origin to the channel center. This procedure ensures that the channel does not need to be precisely centered in the camera's field of view during the measurements. However, the channel should be aligned as precisely as possible with the field of view. To ensure alignment, we recommend to rotate the camera, as opposed to the slide that holds the channels.

## Shear stress profile inside a channel with a square cross-section

The fluid shear stress σ in the mid plane of a channel (blue shading in *Figure 1b*) with length $L$ and square cross section of height $H$ and width $W$ only depends on the radial position $y$ and the total applied pressure $\Delta P$ according to an infinite-series expression (*Delplace, 2018*).

$$\sigma(y) = \left| \frac{4H^2 \Delta P}{\pi^3 L} \sum_{n,\text{odd}}^{\infty} (-1)^{\frac{n-1}{2}} \frac{\pi}{n^2 H} \cos\left(\frac{n\pi z}{H}\right) \frac{\sinh\left(\frac{n\pi y}{H}\right)}{\cosh\left(\frac{n\pi W}{2H}\right)} \right| \tag{4}$$

For all practical purposes, it is sufficient to compute the infinite series for the first 100 terms.

*Equation 4* assumes laminar uniaxial parallel flow and neglects entrance and exit effects, which is justified for a long and narrow channel as used in this study ($L$=5.8 cm, $W = H = 200$ μm). Note that for a given channel geometry and pressure gradient $\Delta P/L$, the shear stress profile $\sigma(y)$ does not depend on the viscosity of the fluid. Equation 4 remains approximately valid also for non-Newtonian e.g. shear-thinning fluids. *Equation 4* predicts that the shear stress is zero in the center of the channel and monotonically increases towards the channel wall (*Figure 1f*).

We take the shear stress $\sigma(y)$ at the cell center $y$ as the average stress acting on the cell. For cells that overstep the channel center, however, the non-monotonic stress profile implies that the average stress can be larger than the stress at the cell center. Therefore, and because cells near the channel center deform and align only marginally, which makes the computation of mechanical properties error-prone, we exclude all cells from further analysis that are closer than one cell radius to the channel center.

## Velocity profile, shear rate profile, and viscosity

The fit function (*Equation 3*) only approximates the true velocity profile, which is sufficient to efficiently and robustly find the channel center. For subsequent computations that require higher precision, we determine the velocity profile by integrating the shear rate. We compute the shear rate $\dot{\gamma}(y)$ as the shear stress σ (*Equation 4*) divided by the viscosity $\eta$ .

$$\dot{\gamma}(y) = \frac{1}{\eta}\sigma \qquad (5)$$

For shear thinning fluids such as alginate solutions, the viscosity $\eta$ is not constant but depends on the shear rate $\dot{\gamma}$. We describe the shear thinning behaviour of the viscosity by the Cross model (*Cross, 1965*)

$$\eta(\dot{\gamma}) = \frac{\eta_0}{1+(\tau\dot{\gamma})^\delta} \qquad (6)$$

with zero-shear viscosity $\eta_0$, relaxation time $\tau$ and power-law shear shear-thinning exponent δ (*Figure 2—figure supplement 2*).

When *Equation 6* is inserted into *Equation 5*, we obtain.

$$\dot{\gamma}(y) = \frac{1+(\tau\dot{\gamma}(y))^\delta}{\eta_0}\sigma \qquad (7)$$

This equation can be written as.

$$0 = \frac{\sigma(y)}{\eta_0} - \dot{\gamma}(y) + \frac{\sigma(y)}{\eta_0}\tau^\delta \cdot \dot{\gamma}(y)^\delta \qquad (8)$$

and numerically solved for $\dot{\gamma}(y)$ by root finding using the Newton-Raphson method.

Finally, to obtain the velocity profile $v(y)$, we integrate the numerically obtained shear rate $\dot{\gamma}(y)$ over the channel, using 5 point Gaussian quadrature

$$v(y) = \int_{W/2}^{y} \dot{\gamma}(y')\, dy' \qquad (9)$$

with the boundary condition $v_{y=W/2} = 0$. The viscosity parameters ($\eta_0$, $\tau$, δ) that best match the velocity profile are determined as follows. We choose five Gaussian quadrature points $y'$ between (0,$W$/2) and numerically compute $\dot{\gamma}$ at the quadrature point $y'$ using *Equation 8*. To ensure convergence, we start iterating with a value of $\dot{\gamma}$ that yields the maximum of the right-hand side of *Equation 8* plus a small number $\epsilon$. The weighted sum of $\dot{\gamma}$ at the Gaussian quadrature points $y'$ is then the velocity at the radial position $y$. This procedure is repeated for different values of ($\eta_0$, $\tau$, δ) until a minimum of the squared differences between the measured and fitted velocity profile is found.

We find that the rheological parameters ($\eta_0$, $\tau$, δ) of the suspension medium obtained this way closely agree with cone-plate rheology measurements (*Müller et al., 2007*). Moreover, the velocity profile for different pressure values can be accurately predicted (*Figure 2—figure supplement 1*), demonstrating that *Equation 6* accurately describes the shear thinning behavior of the suspension fluid.

## Computing the shear strain from the cell shape

Suspended cells under zero shear stress have an approximately circular shape with radius $r_0$. When exposed to constant shear stress, the cell deforms to an elliptical shape with semi-major axis $\tilde{a} = a/r_0$ and semi-minor axes $\tilde{b} = b/r_0$ (in $x, y$-direction) and $\tilde{c} = c/r_0$ (in z-direction), normalized to the radius $r_0$ of the undeformed cell, so that $1 = \tilde{a} \cdot \tilde{b} \cdot \tilde{c}$. Assuming the cell consists of an incompressible material and the stress inside the deformed cell is uniform, the strain $\epsilon$ can be computed from $\tilde{a}$, $\tilde{b}$ and $\tilde{c}$ using (*Equation 10*; *Roscoe, 1967*).

$$\epsilon = (\tilde{a}^2 - \tilde{b}^2)/2I \qquad (10)$$

(corresponding to the right-hand side of Equation 79 in *Roscoe, 1967* without the sign error).

This requires solving a set of shape integrals that depend on the semi-major axis a and semi-minor axis b.

$$I = \frac{2}{5}\frac{g_1'' + g_2''}{g_2'' g_3'' + g_3'' g_1'' + g_1'' g_2''} \qquad (11)$$

$$g_1'' = \int_0^\infty \frac{\lambda}{(\tilde{b}^2+\lambda)(\tilde{c}^2+\lambda)\Delta'} \, d\lambda \tag{12}$$

$$g_2'' = \int_0^\infty \frac{\lambda}{(\tilde{a}^2+\lambda)(\tilde{c}^2+\lambda)\Delta'} \, d\lambda \tag{13}$$

$$g_3'' = \int_0^\infty \frac{\lambda}{(\tilde{a}^2+\lambda)(\tilde{b}^2+\lambda)\Delta'} \, d\lambda \tag{14}$$

With the integration variable $\lambda$ . $\Delta'$ is defined as.

$$\Delta' = \sqrt{(\tilde{a}^2+\lambda)(\tilde{b}^2+\lambda)(\tilde{c}^2+\lambda)}. \tag{15}$$

(*Equation 11* corresponds to Equation 39 in *Roscoe, 1967*, and *Equation 12* corresponds to Equation 18 in *Roscoe, 1967*.)

The shape integral $I$ is pre-computed for different ratios of $\tilde{a}$ and $\tilde{b}$ and then taken from a look-up table.

## Computing the cells' storage and loss modulus

We calculate $G'$ from σ, β, $a$, $b$ according to *Roscoe, 1967*.

$$\frac{5}{2}\frac{\sigma}{G'}\sin(2\beta) = \epsilon(a,b) \tag{16}$$

(corresponding to the left-hand side of Equation 79 in *Roscoe, 1967*).

We calculate $G''$ from β, $\tilde{a}$, $\tilde{b}$, $\eta$ , $\omega$ according to *Roscoe, 1967*.

$$\cos(2\beta) = \left(\frac{\tilde{a}^2-\tilde{b}^2}{\tilde{a}^2+\tilde{b}^2}\right)\frac{1+\frac{2}{5}\frac{\eta-G''/\omega}{\eta}\frac{1}{K}\left(\frac{\tilde{a}^2+\tilde{b}^2}{2\tilde{a}\tilde{b}}\right)^2}{1+\frac{2}{5}\frac{\eta-G''/\omega}{\eta}\frac{1}{K}\left(\frac{\tilde{a}^2-\tilde{b}^2}{2\tilde{a}\tilde{b}}\right)^2} \tag{17}$$

(corresponding to Equation 80 in *Roscoe, 1967*).

with

$$K = \frac{1}{5g_3'}\frac{\tilde{a}^2+\tilde{b}^2}{\tilde{a}^2\tilde{b}^2} \tag{18}$$

(corresponding to Eqation 43 in *Roscoe, 1967*).

$$g_3' = \int_0^\infty \frac{1}{(\tilde{a}^2+\lambda)(\tilde{b}^2+\lambda)\Delta'} \, d\lambda \tag{19}$$

(corresponding to Equation 21 in *Roscoe, 1967*).

A given volume element inside the cell is compressed and elongated twice during a full rotation. Hence, the frequency $\omega$ at which $G'$ and $G''$ is obtained using *Equation 16* and *Equation 17* is twice the angular tank-treading frequency $2\omega_{tt}$.

## Tank treading

We measure the tank-treading frequency as follows. We observe each cell as it travels through the field-of-view and cut-out small image frames with the cell at its center (*Figure 2h*). We then track the movement of characteristic small features using optical flow estimated by the TV-L1 algorithm (*Zach et al., 2007*; *van der Walt et al., 2014*), and calculate their speed and distance during their rotation around the cell's center. The speed versus the ellipse-corrected radius is fitted with a linear relationship to determine the average angular speed (*Figure 2i*). The slope of this relationship is taken as the rotation frequency of the cell.

In cases where the tank-treading frequency cannot be measured (e.g. due to poor contrast or absence of cell-internal features that can be tracked), we estimate the tank-treading frequency following the approach outlined in *Snijkers et al., 2011*. Data shown in *Figure 2j and k* demonstrate that the measured rotation rate $\omega_{tt}/\dot{\gamma}$ (angular frequency divided by the local shear rate) collapses onto a master relationship when plotted against the shear rate. The angular tank-treading frequency $\omega_{tt} = 2\pi f_{tt}$ of the cells can then be predicted with an empirical relationship according to

$$\omega_{\text{tt}}(y) = \frac{\dot{\gamma}(y)}{2} \frac{1}{1 + (0.113 \cdot \dot{\gamma}(y))^{0.45}} \tag{20}$$

when $\dot{\gamma}$ is given in units of 1 /s (*Snijkers et al., 2011*).

## Scaling the rheology

Cells show power-law rheology according to *Equation 1*, which implies that the cell stiffness $k$ and the power-law exponent α (cell fluidity) fully describe the cell rheological properties. Cell stiffness $k$ and cell fluidity α can be obtained from $G'$ and $G''$ by rearranging *Equation 1* as follows

$$k = \frac{G'}{(\omega/\omega_0)^\alpha \, \Gamma(1-\alpha) \cos\left(\frac{\pi}{2}\alpha\right)} \tag{21}$$

$$\tilde{G}(\omega) = \frac{1-V}{4\sqrt{R\delta_0}} \frac{F(\omega)}{\delta(\omega)} - i\omega b(0) \tag{22}$$

with $\omega = 2\omega_{\text{tt}}$ and $\omega_0 = 2\pi$ rad/s. We use a Gaussian kernel density estimation (*Silverman, 1965*; *Virtanen et al., 2020*) to compute the mode of the 2-D distribution for stiffness $k$ and fluidity α, which corresponds to the "most representative" cell with the highest joint probability for stiffness $k$ and fluidity α.

## Software flow-chart

In the following, we summarize the sequence of steps and procedures for measuring cell mechanical properties with our method (*Figure 1—figure supplement 1*).

1. First, typically 10,000 image frames of cells flowing through the channel are recorded with an image acquisition program (recording.py, https://github.com/fabrylab/shear_flow_deforma-tion_cytometer). Second, the images are analyzed off-line with an evaluation pipeline (evaluate.py, https://github.com/fabrylab/shear_flow_deformation_cytometer). The pipeline loads the images and finds and segments cells at the focal plane using a neural network (*Ronneberger et al., 2015*). From the segmented cell shape, morphological properties ($x,y$ position, half major and minor axes $a$ and $b$, orientation β, solidity, circumference) are extracted using the regionprops method of the skimage library (*van der Walt et al., 2014*). Poorly or erroneously segmented cells that deviate from an elliptical shape are filtered out based on circumference and solidity criteria. From a measurement with 10,000 image frames, typically 5000–10,000 cells are identified for subsequent analysis.
   Next, the program identifies cells that are detected across multiple subsequent frames, based on shape and position, computes the flow speed, and applies an particle image velocimetry algorithm to extract the tank treading frequency ftt. *Equation 3* is then fitted to the speed versus y-position relationship of all cells, yielding the channel center yc and the maximum flow speed vmax.
2. The shear stress acting at the center position of each cell is computed using *Equation 4*.
3. The shear rate at the center position of each cell is computed using a set of equations as described above (*Equations 5–9*). This procedure also yields the parameters that describe the viscosity and shear-thinning rheology of the suspension fluid (*Equation 6*).
4. The cell strain is computed from the half major and minor axis $a$ and $b$ using *Equation 10*. Subsequently, $G'$ and $G''$ of each cell at twice its angular tank treading frequency is computed using *Equation 16* and *Equation 17*.
5. To compare the mechanical properties of cells that have experienced different tank-treading frequencies, we scale $G'$ and $G''$ to a frequency of 1 Hz using *Equation 22* and *Equation 21*, yielding the stiffness $k$ and fluidity α of individual cells. The average stiffness $k$ and fluidity α of the cell population is determined from the maximum of the two-dimensional Gaussian kernel density computed using the scipy.stats.gaussian_kde method of the scipy library (*Silverman, 1965*; *Virtanen et al., 2020*).

## PAAm reference bead preparation

Polyacrylamide hydrogel microparticles (PAAm beads) are produced using a flow-focusing PDMS-based microfluidic chip described in *Girardo et al., 2018*. Briefly, a stream of a polyacrylamide pre-gel mixture is squeezed by two counter-flowing streams of an oil solution to form droplets with a mean diameter in the range of 11.5–12.5 μm. The oil solution is prepared by dissolving ammonium Krytox surfactant (1.5% w/w), N,N,N′,N′-tetramethylethylenediamine (0.4% v/v), and acrylic

acid N-hydroxysuccinimide ester (0.1% w/v) in hydrofluoroether HFE 7500 (Ionic Liquid Technology, Germany). The pre-gel mixture is obtained by dissolving and mixing acrylamide (40% w/w), bis-acrylamide (2% w/w) and ammonium persulfate (0.05% w/v) (all from Merck, Germany) in 10 mM Tris-buffer (pH 7.48). Particles with three different elasticities are obtained by diluting the pre-gel mixture in Tris-buffer to final acrylamide-bisacrylamide concentrations of 3.9%, 5.9%, 6.9% respectively. Alexa Fluor 488 Hydrazide (ThermoFisher Scientific, Germany) is dissolved in D.I. water (stock solution 3 mg/ml) and added to the mixture for a final concentration of 55 µg/ml to make the particles fluorescent. Droplet gelation is carried out at 65 °C for 12 hr. The droplets are washed and resuspended in 1 x PBS.

## Atomic force microscopy (AFM) of cells and PAAm beads

AFM-based microrheology measurements for PAAm beads are performed using a Nanowizard 4 (JPK BioAFM, Bruker Nano GmbH, Berlin). The measurements are carried out using a wedged cantilever with a flat surface parallel to the measurement dish. The cantilever is prepared by applying a UV curing glue to a tipless cantilever (PNP-TR-TL, nominal spring constant $k = 0.08$ N/m used for the stiff (6.9% $C_{AAmBis}$) beads, or Nanoworld or Arrow-TL1, nominal spring constant $k = 0.03$ N/m used for the medium (5.9% $C_{AAmBis}$) and soft (3.9% $C_{AAmBis}$) beads as described in *Stewart et al., 2013*. Prior to each experiment, the optical lever sensitivity is measured from the force-distance relationship of a polystyrene bead attached to a glass surface, and the cantilever spring constant is measured using the thermal noise method (*Hutter and Bechhoefer, 1993*). Measured spring constants are 0.09 N/m for PNP-TR-TL cantilevers, and 0.018 N/m for Arrow-TL1 cantilevers.

To perform the AFM microrheology measurements, the cantilever is lowered with a speed of 10 µm/s until a force between 1–3 nN is reached, corresponding to an indentation depth $\delta_0$ between 1.5–3 µm. The cantilever is then sinusoidally oscillated with an amplitude of 30 nm for a period of 10 cycles. This procedure is repeated for different oscillation frequencies in the range between 0.1–150 Hz. To extract the complex shear modulus $G^*$ of the PAAm beads, the force-indentation curves are analyzed as described in *Alcaraz et al., 2003* using the Hertz model that describes the deformation of a soft sphere between two flat surfaces in the limit of small deformations. The complex shear modulus is then computed according to

$$\tilde{G}(\omega) = \frac{1-V}{4\sqrt{R\delta_0}} \frac{F(\omega)}{\delta(\omega)} - i\omega b(0) \tag{23}$$

where $\nu$ is the Poisson ratio of the PAAm bead (assumed to be 0.5), $\omega$ is the angular frequency of the oscillations, $F(\omega)$ and $d(\omega)$ are the Fourier transforms of the force and indentation signal, $R$ is the radius of the PAAm bead, $\delta_0$ is the initial indentation, and $b(0)$ is the hydrodynamic drag coefficient of the cantilever with the surrounding liquid. The hydrodynamic drag coefficient is measured as described in *Alcaraz et al., 2002* and estimated to be $b(0) = 5.28$ Ns/m for PNP-TR-TL cantilevers and $b(0) = 29.7$ Ns/m for Arrow TL1 cantilevers.

AFM-based measurements for THP1 cells are performed with four-sided regular pyramidal-tipped MLCT-bio-DC(D) cantilevers (Bruker). The spring constant of the cantilever is measured from the thermal noise spectrum in air, and the optical lever sensitivity is measured from the thermal noise spectrum in liquid (*Sumbul et al., 2020*). The cells are immobilized to plastic petri dishes coated with poly-L-lysine at a concentration of 0.01 mg/mL for 10 min. Force curves are measured at 3 or more positions around the cell center for a constant indentation speed of 5 µm/s up to a maximum force of 0.8 Nn. At each position, at least 3 force-distance curves are obtained. We determine the viscoelastic step-response stress relaxation function $E(t)$ of the cell by least-square fitting the theoretical force response to the measured force curve during indentation with a pyramidal tip (*Efremov et al., 2017*)

$$F(t,\delta(t)) = \begin{cases} \frac{3\tan\theta}{4(1-\nu^2)} \int_0^t E(t-\tau)\frac{\partial\delta^2}{\partial\tau}d\tau, & 0 \le t \le t_m \\ \frac{3\tan\theta}{4(1-\nu^2)} \int_0^{t_1(t)} E(t-\tau)\frac{\partial\delta^2}{\partial\tau}d\tau, & t_m \le t \le t_{ind} \end{cases} \tag{24}$$

where $F$ is the force acting on the cantilever tip, δ is the indentation depth, $t$ is the time since initial contact, $t_m$ is the duration of approach phase, $t_{ind}$ is the duration of complete indentation cycle, and $t_1$ is the auxiliary function determined by the equation

$$\int_{t_1(t)}^t E(t-\tau)\frac{\partial\delta}{\partial\tau}d\tau = 0 \tag{25}$$

The viscoelastic step response function $E(t)$ is assumed to follow the relationship

$$E(t) = 2k(1 + \nu) \left( \frac{2\pi t}{t_0} \right)^{-\alpha} \tag{26}$$

where the reference time $t_0$ is set to 1 s so that $k$ is the cell's shear modulus measured at time $t = 0.159$ s (corresponding to $\omega=1$ rad/s as in the flow deformability measurements). The cell's Poisson ratio $\nu$ is assumed to be 0.5, and α is the cell's fluidity.

## Rheology of alginate solutions

We measure the viscosity of the alginate solution at a temperature of 25 °C at shear rates between 0.01 s$^{-1}$ and 1000 s$^{-1}$ using a temperature-controlled rheometer (DHR-3, TA-Instruments, USA) with stainless steel cone and plate (diameter of 40 mm with a cone angle of 2° and a 65 µm truncation gap). Temperature is controlled with a Peltier-element. Equilibration time and measurement time are set to 30 seconds for every measurement point (logarithmic sweep, 5 points per decade). Every sample is rested for three minutes inside the rheometer to ensure temperature equilibration. A solvent trap with deionized water is used to prevent drying of the alginate samples.

## Cell cycle measurement with Fucci

We use NIH-3T3 cells that display the fluorescent ubiquitination-based cell cycle indicator (Fast-FUCCI) reporter system after lentiviral transduction. The lentivirus is generated by transfection of Lenti-X 293T cells (Takara, #632180) with pBOB-EF1-FastFUCCI-Puro (Addgene, #86849), a packaging plasmid psPAX2 (Addgene, #12260), and an envelope plasmid pCMV-VSV-G (Addgene, #8454), using Lipofectamine 2000 reagent (Invitrogen, #11668–019). 48 hr after transfection, infectious lentivirus-containing supernatant is harvested, centrifuged (500 x g, 10 min), and 10-fold concentrated using the Lenti-X-concentrator reagent (Takara, #631232). NIH-3T3 cells are seeded 24 hr prior to transduction at a density of 10 000 per cm$^2$. Three days after transduction, cells are cultured for at least 5 additional days in medium containing puromycin (5 µg/ml) to select successfully transduced cells.

In our shear flow deformation cytometry setup, we measure only the green fluorescence signal, indicating cells in S, G2 and early M-phase (*Sakaue-Sawano et al., 2008*), and deduce that cells with a green fluorescence intensity below a certain threshold are in G1 phase. To set this threshold, we measure both the red fluorescence signal (indicating cells in G1 phase *Sakaue-Sawano et al., 2008*) and the green fluorescence signal of individual cells prior to harvesting, using an epifluorescence microscope. We then compute the green-fluorescence intensity threshold, normalized to the median intensity that best separates the cells in G1 phase from the cells in S, G2 and early M-phase. Because some cells fluoresce green and red at the same time, 22.6% of cells in G1 phase and 2.4% of the cells in S, G2 and early M-phase are erroneously classified when the classification is based on the green fluorescence signal alone. After harvesting and suspending the cells in alginate, they are measured in the shear flow setup. Bright-field images are analyzed as described above to segment cells that are in focus, and the fluorescence intensities are averaged over the segmented cell area.

## Acknowledgements

This study was supported by the Deutsche Forschungsgemeinschaft (TRR-SFB 225 project 326998133 subprojects A01, A07 and B07), and the European Union's Horizon 2020 research and innovation programmes No 812772 (project Phys2BioMed, Marie Skłodowska-Curie grant) and No 953121 (project FLAMIN-GO). We thank Jonas Hazur and Aldo Boccaccini for helpful discussions and for providing the alginate.

## Additional information

### Competing interests

Richard Gerum: is inventor in a patent application on this method (EP22150396.4) alongside SG and BF. Stephan Gekle, Ben Fabry: is an inventor in a patent application on this method (EP22150396.4). The other authors declare that no competing interests exist.

## Funding

| Funder | Grant reference number | Author |
|---|---|---|
| Deutsche Forschungsgemeinschaft | TRR-SFB 225 subprojects A01 | Elham Mirzahossein |
| Horizon 2020 | No 812772 | Mar Eroles |
| Horizon 2020 | No 953121 | Mar Eroles |
| Deutsche Forschungsgemeinschaft | B07 | Stephan Gekle |
| Deutsche Forschungsgemeinschaft | A07 | Stefan Schrüfer |

The funders had no role in study design, data collection and interpretation, or the decision to submit the work for publication.

## Author contributions

Richard Gerum, Software, Formal analysis, Supervision, Validation, Investigation, Visualization, Methodology, Writing – original draft, Writing – review and editing; Elham Mirzahossein, Software, Formal analysis, Investigation, Methodology; Mar Eroles, Formal analysis, Investigation, Methodology; Jennifer Elsterer, Data curation, Formal analysis, Investigation, Methodology; Astrid Mainka, Shada Abuhattum, Ruchi Goswami, Stefan Schrüfer, Dorothea Schultheis, Investigation, Methodology; Andreas Bauer, Selina Sonntag, Alexander Winterl, Johannes Bartl, Software, Methodology; Lena Fischer, Resources, Investigation, Methodology; Salvatore Girardo, Supervision, Writing – review and editing; Jochen Guck, Writing – review and editing; Nadine Ströhlein, Mojtaba Nosratlo, Software, Investigation, Methodology; Harald Herrmann, Resources, Investigation, Methodology, Writing – review and editing; Felix Rico, Supervision, Investigation, Methodology, Writing – original draft; Sebastian Johannes Müller, Software, Formal analysis, Methodology; Stephan Gekle, Formal analysis, Supervision, Methodology; Ben Fabry, Conceptualization, Formal analysis, Supervision, Visualization, Methodology, Writing – original draft, Project administration, Writing – review and editing

## Author ORCIDs

Richard Gerum (ID) http://orcid.org/0000-0001-5893-2650
Mar Eroles (ID) http://orcid.org/0000-0003-3571-0769
Felix Rico (ID) http://orcid.org/0000-0002-7757-8340
Sebastian Johannes Müller (ID) http://orcid.org/0000-0002-6020-4991
Ben Fabry (ID) http://orcid.org/0000-0003-1737-0465

## Decision letter and Author response

Decision letter https://doi.org/10.7554/eLife.78823.sa1
Author response https://doi.org/10.7554/eLife.78823.sa2

# Additional files

## Supplementary files
• MDAR checklist

## Data availability
Software is made available at GitHub, https://github.com/fabrylab/shear_flow_deformation_cytometer, (copy archived at swh:1:rev:2d35a697243c432cddd52e10d2e3c5121f922adf). CSV files containing the data of all individual cells used for the study have been made available on Dryad (https://doi.org/10.5061/dryad.5hqbzkh8p).

The following dataset was generated:

| Author(s) | Year | Dataset title | Dataset URL | Database and Identifier |
|---|---|---|---|---|
| Gerum R, Mirzahossein E, Eroles M, Elsterer J, Mainka A, Bauer A, Sonntag S, Winterl A, Bartl J, Fischer L, Abuhattum S, Goswami R, Girardo S, Guck J, Schrüfer S, Ströhlein N, Nosratlo M, Herrmann H, Schultheis D, Rico F, Müller SJ, Gekle S, Fabry B | 2022 | Viscoelastic properties of suspended cells measured with shear flow deformation cytometry | https://doi.org/10.5061/dryad.5hqbzkh8p | Dryad Digital Repository, 10.5061/dryad.5hqbzkh8p |

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
