## [Editor Report]

This paper describes an inexpensive but very powerful microfluidic approach to quantitatively determine the viscoelastic properties of living cells from their deformation in a flow. Its implementation seems simple so that even people not specialized in cell mechanics can use it, and the method offers the possibility to perform measurements on a large number of cells (up to 50-100 per second). The data are compelling and this technique should set a new standard in the field.

---

## [Decision Letter]

**Decision letter after peer review:**

Thank you for submitting your article "Viscoelastic properties of suspended cells measured with shear flow deformation cytometry" for consideration by *eLife*. Your article has been reviewed by 2 peer reviewers, and the evaluation has been overseen by a Reviewing Editor and Anna Akhmanova as the Senior Editor. The following individuals involved in review of your submission have agreed to reveal their identity: Clément Campillo (Reviewer #1); Timo Betz (Reviewer #2).

Essential revisions:

You will see from the comments of the two Reviewers that they agree on the usefulness of your method. However, they also point out some weaknesses in this study that need to be addressed in the revision. Essential revisions should include:

1) A much more in-depth introduction and discussion of existing methods (see Reviewer 1's comment). The interest of a method paper is certainly to describe new protocols/tools, but also to discuss their interest (and also their disadvantages) compared to existing tools. This is particularly important as papers using microfluidics to characterize the mechanical properties of cells have recently been published (including the paper by Oliver Otto which is mentioned). This discussion would be useful for less specialized readers (e.g. cell biologists wanting to characterize their cells but not necessarily having a strong biophysics background). Why would it make a difference in their experiments to have a frequency-dependent response? In what specific cases would having frequency dependent values allow them to discriminate cells better than static parameters?

These modifications to the manuscript should be particularly easy to make.

2) Further analysis of the experimental data to justify or refute the validity of a single powerlaw approach versus a two powerlaw approach, or a modification of the experimental setup to acquire and present data only in the linear regime (see reviewer comment 2). This would answer two questions:

(a) whether the discrepancy is truly due to strain stiffening, and

b) whether one can reliably use the data in the high frequency level to obtain the correct stiffness and powerlaw exponent. This point will certainly give you much more work, but it is essential to address it to convince us that the final values and model are reliable.

3) Presenting experiments investigating the role of intermediate filaments from the same cell line so that the results are easier to interpret.

*Reviewer #1 (Recommendations for the authors):*

I am not able to evaluate the validity of the Roscoe model and the equations 10 -19, nevertheless the agreement between the measurements obtained by the two techniques used are very convincing. This is even surprising in the case of THP1 cells (Figure 5a and b), because SF-DC and AFM probe cell mechanics at very different scales.

The novelty of the technique compared to Fregin NatComm 2019 has to be discussed in detail as the main claim of the article is the approach for cell mechanics measurements. From a physics point of view, the technique presented here gives for instance G' and G' as a function of frequency, whereas Fregin et al., give only effective cell elasticity and viscosity. Therefore, the measurement of cell's mechanics is more complete with this assay, as described in the manuscript l.217. Note that the experimental setup presented in the manuscript is simpler than the one presented by Fregin et al., which might favor its use by other labs. This justifies the publication of the article in *eLife*, as physical methods are in the scope of the journal. On the other hand, such a detailed physical description might not be required to only discriminate between cell populations, as it is the main application of this type of experiment.

The introduction is very short and the authors start presenting their technique immediately. I think the readers need more context on why these type of measurements are useful, which experimental techniques are used and the mechanical models on which these techniques rely. Some of these aspects are in the Discussion section, to highlight the interest of their technique but the introduction should be more detailed.

On the tank treading movement of cells in shear flows, several articles are cited but the article comports no discussion at all on this mechanism, and how it has been characterized for red blood cells for instance.

Similarly, some readers may not be familiar with viscoelastic models, the article is not very pedagogic on this.

*Reviewer #2 (Recommendations for the authors):*

While I really like the method, and think it should be published in *eLife* after successful revision, I am worried about the reliability of some parts of the paper. Mainly, I think the strain stiffening explanation needs to be better nailed. Why is it not possible to change the conditions, so that the deformation remains in the linear regime throughout the measurement. Even if correct, all the data that is in the strain stiffening regime would then lead to wrong stiffness and powerlaw exponent. Would it not make more sense, to only focus on the results as close to the measurements as possible? We have a G' and G' for a given frequency, why do we need to translate this into other parameters that are even less reliable?

Besides this main concern, I have a couple of other points that I would like to transmit to the authors to consider for a further improvement of the paper.

– Why is in figure 2a/b no data given for the position close to 0? I did not realize an explanation for this when I read the paper. Is it problematic in this regime to determine the deformation?

– In figure 2 g it is shown that for a large enough shear rate the rheometer and the flow cytometer values of viscosity are the same. If I understand this correctly, it also means that only right at the center the shear rate is so low that this matters… however in the distances that correspond to this shear no measurements are given. Is this right? If so, maybe it is worthwhile to mention this in more detail.

– Why seem the stiffness values of the PAA beads to be so pressure dependent as seen in figure 4e. And in figure S3, the values of the AFM and the flow cytometer should be plotted on the same y scale, or ideally even plotted over each other. What keeps you from adding the values a in b,c, and the same for d and g?

– In Equation 2 µ seems not to be defined.

– For the cell experiments with vimentin it is very confusing why you compare WT 3T3 with MEF desmin-knockouts and knockins of vimentin. It looks like you don't have WT MEF. Why comparing different cell types. This makes it hard to believe that your conclusion is correct. At least all differences between 3T3 and MEFs might be because of the different cell type. Use also MEFs for the 'normal' situation.

– Also, there is a statistical test in figure 7 b and c missing. How many cells did you measure for each of the datapoints in 7b,c?

– In the discussion you mention a local viscosity of the fluid. This is a bit misleading as it suggests that the viscosity depends on the position, but it depends on the shear rate (which depends on the position for a given pressure).

– It should be more focused on the actual measurement of G' G' at a single frequency. I think we don't know sufficiently well if a single powerlaw can explain the measurements, to use this hypothesis in a way that makes the reader think the method provides information about frequency dependence from a single snapshot.

– It would be good to get a reference for the statement that you find similar values for the E50 as other authors (line 269).

– Please tone down the statement of line 284-286. There is no statistical test, and you compare apples with oranges… don't say that you have shown the effect of vimentin. All you see is the increase in stiffness in the knock-in situation.

– It would be interesting to provide a measure of variance in the text describing the different \σ values. (line 502-503)

– Why not using a non-shear thinning medium to do the measurements? Everything should be much more simple there.

– You treat \eta_0, \tau, and \δ as independent parameters. Is this really the case, or could some depend on each other? Also please give the values you measured for these parameters.

– You say that stress stiffening happens, but then you use a model (equations 21,22) that breaks down under these conditions. Here you basically say that the measurements you obtain in the high frequency regime do not work. Why are these not excluded, and how do you determine the stress at which we cannot take the G' and G' as unaffected.

– In Line 580 you say the AFM goes from 0.1 to 150Hz, but the measurements you show are up to 10000 rad/s. What is right?

---

## [Author Response]

Essential revisions:You will see from the comments of the two Reviewers that they agree on the usefulness of your method. However, they also point out some weaknesses in this study that need to be addressed in the revision. Essential revisions should include:1) A much more in-depth introduction and discussion of existing methods (see Reviewer 1's comment). The interest of a method paper is certainly to describe new protocols/tools, but also to discuss their interest (and also their disadvantages) compared to existing tools. This is particularly important as papers using microfluidics to characterize the mechanical properties of cells have recently been published (including the paper by Oliver Otto which is mentioned). This discussion would be useful for less specialized readers (e.g. cell biologists wanting to characterize their cells but not necessarily having a strong biophysics background). Why would it make a difference in their experiments to have a frequency-dependent response? In what specific cases would having frequency dependent values allow them to discriminate cells better than static parameters?These modifications to the manuscript should be particularly easy to make.

We have followed these recommendations and have now explained in the Introduction why it is important to measure the frequency-dependent properties (namely to compare measurements across different platforms and time/frequency scales).

2) Further analysis of the experimental data to justify or refute the validity of a single powerlaw approach versus a two powerlaw approach, or a modification of the experimental setup to acquire and present data only in the linear regime (see reviewer comment 2). This would answer two questions:(a) whether the discrepancy is truly due to strain stiffening, andb) whether one can reliably use the data in the high frequency level to obtain the correct stiffness and powerlaw exponent. This point will certainly give you much more work, but it is essential to address it to convince us that the final values and model are reliable.

As we explain in more detail in our response to the reviewers, we erroneously assumed that the probing frequency equals the tank treading frequency. However, the probing frequency is twice the tank-treading frequency. With this correction, a single power-law now describes our data much better. We have re-analyzed all data with this correction, and moreover we have changed the reference frequency (for the cell stiffness k) from 1 rad/s now to 1 Hz, which we think is more intuitive for most readers.

Further, we actually have used a two-power law description of cell mechanics already in our original submission (Equation 2), and we use a single power law description only where the frequency or shear rate of the measurements is so low that the second power law does not yet come into play, which is the case for all our shear flow deformation cytometry measurements on cells.

Our approach of describing the frequency response with a power law (Equations 1 and 2) has been validated by numerous different techniques, in vastly different cells, and by many different laboratories. It has been shown to be valid also for strain-stiffening cells, it correctly models the frequency response of cells over a very large range of frequencies (spanning at least 5 orders in magnitude), and it allows for the comparison of data obtained at different frequencies or shear rates (which most other microfluidic techniques do not). We have provided more detailed explanations below.

3) Presenting experiments investigating the role of intermediate filaments from the same cell line so that the results are easier to interpret.

We actually have used the same cell type, a fibroblast line derived from mouse embryo fibroblasts (MEF) by spontaneous immortalization. It is important to note that the NIH-3T3 cell line is derived from MEFs just as our Vim (-/-) cell line. We had stated this previously only in Methods, now explicitly state this also in the main text in Results.

Reviewer #1 (Recommendations for the authors):I am not able to evaluate the validity of the Roscoe model and the equations 10 -19, nevertheless the agreement between the measurements obtained by the two techniques used are very convincing. This is even surprising in the case of THP1 cells (Figure 5a and b), because SF-DC and AFM probe cell mechanics at very different scales.

As we explain below in more detail in our answer to Reviewer #2, we had assumed that the probing frequency is the tank-treading frequency. However, the cells are actually probed at twice the tank-treading frequency. After correcting this, we find that measurements with our shear flow deformation cytometer show a higher fluidity compared to AFM measurements and also give a lower stiffness. With other words, the two techniques do not (any longer) give nearly identical absolute values (although they are reasonably similar), but we still find common cell behavior with both techniques, namely (1) power-law rheology, (2) a log-normal distribution of G’ and a normal distribution for α, and (3) an inverse relationship between G’ and α for individual cells. Moreover, in our AFM measurements we apply power-law rheology to individual cells, whereas with our microfluidic technique, we probe each cell only at a single frequency and therefore cannot directly demonstrate that individual cells show that same power-law behavior as the cell population. For these reasons, we still think that the comparison with AFM supports the validity of our approach and method.

The novelty of the technique compared to Fregin NatComm 2019 has to be discussed in detail as the main claim of the article is the approach for cell mechanics measurements.

This was also asked by the editor when we first submitted our manuscript, and we explained the differences to the method of Fregin and the novelty/advantages of our method at various places in Introduction and Discussion. These additions are highlighted in the manuscript in red.

From a physics point of view, the technique presented here gives for instance G' and G' as a function of frequency, whereas Fregin et al., give only effective cell elasticity and viscosity. Therefore, the measurement of cell's mechanics is more complete with this assay, as described in the manuscript l.217. Note that the experimental setup presented in the manuscript is simpler than the one presented by Fregin et al., which might favor its use by other labs. This justifies the publication of the article in eLife, as physical methods are in the scope of the journal. On the other hand, such a detailed physical description might not be required to only discriminate between cell populations, as it is the main application of this type of experiment.The introduction is very short and the authors start presenting their technique immediately. I think the readers need more context on why these type of measurements are useful, which experimental techniques are used and the mechanical models on which these techniques rely. Some of these aspects are in the Discussion section, to highlight the interest of their technique but the introduction should be more detailed.

We followed the suggestion, and in Introduction we now briefly discuss two relevant models (the Kelvin-Voight model with constant parameters, as used by Fregin et al., and the structural damping model that we favor). While this is not meant to represent a comprehensive discussion of models, we hope it helps the reader to better understand what follows. We also added an example that we hope convincingly illustrates how cell rheology measurements might be interesting and useful even for potential readers outside the mechanobiology community.

On the tank treading movement of cells in shear flows, several articles are cited but the article comports no discussion at all on this mechanism, and how it has been characterized for red blood cells for instance.

We expanded our Introduction to explain the mechanism, included two more references, and also expanded our discussion on tank-treading.

Similarly, some readers may not be familiar with viscoelastic models, the article is not very pedagogic on this.

We also expanded the Introduction in this regard and have introduced the Kelvin-Voigt model and the structural damping model, both of which are relevant for what follows.

Reviewer #2 (Recommendations for the authors):While I really like the method, and think it should be published in eLife after successful revision, I am worried about the reliability of some parts of the paper. Mainly, I think the strain stiffening explanation needs to be better nailed. Why is it not possible to change the conditions, so that the deformation remains in the linear regime throughout the measurement. Even if correct, all the data that is in the strain stiffening regime would then lead to wrong stiffness and powerlaw exponent.

Please see our more detailed explanations below where we argue that our method provides a secant modulus and not a small amplitude differential modulus, but that does not make the measurements or numbers “wrong”. Regarding the existence of a “linear regime”, we have addressed this in several previous studies (Kollmannsberger et al., Nonlinear viscoelasticity of adherent cells is controlled by cytoskeletal tension. Soft Matter, 2011 and Lange et al., Unbiased high-precision cell mechanical measurements with microconstrictions. Biophys J, 2017, Kah et al., High-Force Magnetic Tweezers with Hysteresis-Free Force Feedback, Biophys J 2020). The insight from these studies is that non-linearities can start to appear even at small amplitudes of strain or stress.

Would it not make more sense, to only focus on the results as close to the measurements as possible? We have a G' and G' for a given frequency, why do we need to translate this into other parameters that are even less reliable?

It is possible to only report the “raw” G’ and G” of each cell, which is in essence what most other microfluidic cell deformability measurements provide. Our data do not necessarily need to be translated into other parameters (such as stiffness k and fluidity α) if one wants to compare different measurements obtained at similar tank-treading frequencies or time scales. However, as explained above, converting G’ and G” into frequency-independent stiffness and fluidity values makes it possible to compare our measurements to other data obtained with other techniques that operate at different, non-overlapping time- or frequency-scales.

Besides this main concern, I have a couple of other points that I would like to transmit to the authors to consider for a further improvement of the paper.– Why is in figure 2a/b no data given for the position close to 0? I did not realize an explanation for this when I read the paper. Is it problematic in this regime to determine the deformation?

At low shear stress, when cell deformations are small and cells appear nearly circular, it is difficult to reliably measure the alignment angle. Errors in the alignment angle affect the accuracy of our G” estimate. Furthermore, when cells in the center of the microfluidic channel are exposed to a non-monotonic shear stress profile, they can deform (depending on their stiffness) into a bullet-like shape, which Roscoe theory does not consider. Therefore, we only include cells that experience a monotonic shear stress, i.e., that are completely on either side of the channel center. We have included this explanation in Methods.

– In figure 2 g it is shown that for a large enough shear rate the rheometer and the flow cytometer values of viscosity are the same. If I understand this correctly, it also means that only right at the center the shear rate is so low that this matters…

Actually, the data shown in Figure 2g demonstrates a good correspondence of the viscosity values between a cone-plate rheometer and our device not only for high shear rates but also for low shear rates (e.g., below 10/s). Regarding the position within the microfluidic channel where shear thinning sets in: this depends on the applied pressure. At lower pressures (e.g., 0.5 bar, see Figure 2e), the shear rate of the suspension fluid remains below 100/s, and hence most of the cells do not experience the shear thinning regime. At 3 bar, by contrast, almost all cells are well in the shear-thinning regime of the suspension medium except for those close to the center of the channel. For Roscoe theory, however, it does not matter if the suspension fluid is shear thinning or not, as long as we know the local value of the viscosity, which we do.

however in the distances that correspond to this shear no measurements are given. Is this right? If so, maybe it is worthwhile to mention this in more detail.

Correct, we do not evaluate cells at or near the center of the channel. The reason, however, is not that we cannot accurately determine the shear rate and viscosity at this location. Rather, the reason is the non-monotonic shear stress profile, as explained above.

– Why seem the stiffness values of the PAA beads to be so pressure dependent as seen in figure 4e.

Actually, the stiffness-values we find for PAA beads are nearly pressure (and strain) independent, as expected for a linear material such as PAA (Figure 4e). However, we find that the power law exponent, which is very low for a nearly perfect elastic material such as PAA, tends to increase at large strains, presumably due to poro-elastic effects (Figure 4f). Put simply, the PAA polymer network interacts with the background fluid of the PAA hydrogel, similar to the cytoskeleton that is dragged through the cytoplasm when cells are deformed. Hence the viscosity term in Equation 2 captures the high frequency rheology of both, cells and PAA beads.

And in figure S3, the values of the AFM and the flow cytometer should be plotted on the same y scale, or ideally even plotted over each other. What keeps you from adding the values a in b,c, and the same for d and g?

We followed this suggestion and now plot the AFM and flow cytometer data with the same y-scale.

– In Equation 2 µ seems not to be defined.– For the cell experiments with vimentin it is very confusing why you compare WT 3T3 with MEF desmin-knockouts and knockins of vimentin. It looks like you don't have WT MEF. Why comparing different cell types. This makes it hard to believe that your conclusion is correct. At least all differences between 3T3 and MEFs might be because of the different cell type. Use also MEFs for the 'normal' situation.

The NIH-3T3 cell line has been established from mouse embryonic fibroblasts (MEFs) by spontaneous immortalization (Todaro and Green (1963) Quantitative studies of the growth of mouse embryo cells in culture and their development into established lines. J Cell Biol 17, 299-313). We followed the corresponding protocol for MEFs obtained from vimentin-knockout mouse embryos (Golucci-Guyon et al., (1994) Mice lacking vimentin develop and reproduce without an obvious phenotype. Cell 79, 679-694). The Vim(-/-)Des cell line was generated by stable transfection of the Vim(-/-) cell lines with desmin. Thus, the three cell lines are of the same cell type. This information was given in Methods; we now have also added an explanation in Results.

– Also, there is a statistical test in figure 7 b and c missing. How many cells did you measure for each of the datapoints in 7b,c?

The minimum number of cells was 313, the maximum number was 4884, with an average of 1813 cells per data point and a standard deviation of 1191. The average number is now specified in the figure legend. We also performed a statistical analysis and now show (Figure 7c) that the cytochalasin-D response for the vin(-/-) cells is significantly increased compared to control, whereas the cytochalasin-D responses between control and the desmin knock-in cells are not significantly different.

– In the discussion you mention a local viscosity of the fluid. This is a bit misleading as it suggests that the viscosity depends on the position, but it depends on the shear rate (which depends on the position for a given pressure).

To clarify, we added “shear dependent” (local shear dependent viscosity).

– It should be more focused on the actual measurement of G' G' at a single frequency. I think we don't know sufficiently well if a single powerlaw can explain the measurements, to use this hypothesis in a way that makes the reader think the method provides information about frequency dependence from a single snapshot.

We agree. In the revised Introduction, we clarify that Roscoe theory makes no predictions about the frequency-dependence of G’ and G” for that particular cell. However, we find a power-law in the G’ and G” vs frequency relationship of the ensemble, and this suggests that the G’ and G” vs frequency relationship of (most of the) individual cells also follow a power-law.

– It would be good to get a reference for the statement that you find similar values for the E50 as other authors (line 269).

We refer to Urbanska et al., Nat Meth 2020.

– Please tone down the statement of line 284-286. There is no statistical test, and you compare apples with oranges… don't say that you have shown the effect of vimentin. All you see is the increase in stiffness in the knock-in situation.

Regarding the apples and oranges, see our explanations above (all measurements, including those for wild-type cells, were performed with mouse embryonic fibroblast-derived cell lines). Besides, we see more than just an increase in stiffness in the knock-in situation, namely we find a decrease in the relative effects of actin cytoskeleton depolymerization. In our manuscript, we mainly focus on the latter observation, because such relative changes are thought to be less sensitive towards the different life histories of the WT MEF, Vim ko MEF and Des ki MEF cell lines. Nonetheless, also the fact that both the WT and Des ki MEFs are stiffer than the Vim ko MEFs supports our assertion that “the stable introduction of a cytoplasmic intermediate filament protein into intermediate filament-free cells restores their cytoskeletal functionality and mechanical stability”. We also added a statistical test Mann Whitney-U, which supports this statement (see Figure 7c).

– It would be interesting to provide a measure of variance in the text describing the different \σ values. (line 502-503)

The rather convoluted way of computing the acting shear stress (from the average stress at the extreme points) originated from the time when we still included cells that overlapped with the channel center. There, the shear stress at the cell center can reach zero, but a cell will still deform as the average shear stress is non-zero and approximately the average stress at the extreme points. But since we now exclude all cells that overstep the channel center, we directly use the shear stress at the cell center instead (the deviation between the two ways to calculate the acting stress, however, is less than 1% for most cells). This is now explained in Methods, together with an explanation why we exclude cells near the channel center. We have re-evaluated all data accordingly.

– Why not using a non-shear thinning medium to do the measurements? Everything should be much more simple there.

The shear-thinning behavior complicates matters only moderately, since we obtain the local viscosity (which is important only for the calculation of G") as a side result of our method. Nonetheless, we tried glycerin, but to achieve sufficiently high viscosities, the water content needs to be low, and the cells appeared to suffer. Methyl cellulose or xanthan gum solutions, two other commonly used cell suspension media which we tried and which can be used, are also shear thinning.

– You treat \eta_0, \tau, and \δ as independent parameters. Is this really the case, or could some depend on each other? Also please give the values you measured for these parameters.

We now show the parameters of eta_0, tau, and δ for different alginate concentrations in Figure 2-supplementary figure 2. Regarding the question of the independence or covariance of the parameters – this is difficult to determine in a mathematically clean way as we extract these parameters from a fitting routine that iteratively reduces the error between the measured and modeled flow profil, but there are numerous intermediate calculations involved such as root finding and numerical integration. But this is what we can say:

eta_0 is the low shear stress viscosity and as such determines most sensitively the flow speed at the channel center. By contrast, δ describes the shear thinning behavior at large shear rates and determines the overall shape of the flow curve (which approaches a nearly quadratic function for a Newtonian fluid (δ = 0) and a plug-flow-like profile for higher δ values). Because of their distinct effects, eta_0 and δ show a low covariance

Tau sets the shear rate at which the viscosity behavior transits from a Newtonian regime to a shear-thinning regime. The value of tau affects both the magnitude and the shape of the velocity profile and therefore shows a higher covariance with both eta and δ. Between consecutive measurements of the same fluid, tau shows the largest relative fluctuations, whereas eta_0 and δ are more stable (the data points in Figure 2-supplementary figure 2 represent individual measurements, some of it performed on the same day, i.e. with the same alginate preparation, but also on different days, i.e. each time with new alginate preparations of the same concentration).

– You say that stress stiffening happens, but then you use a model (equations 21,22) that breaks down under these conditions.

In fact, Roscoe theory considers a neo-Hookean material, which is strain stiffening. Hence, Roscoe theory does not break down in the case of a stress- or strain-stiffening material (and neither does power law rheology, see for example Kollmannsberger Soft Matter 2011 or Lange BJ 2017). However, the strain stiffening of cells appears to be more pronounced than the stiffening predicted for a neo-Hookean material. Therefore, our measurements represent an effective secant modulus and not a small-strain tangential modulus, but nonetheless, also the G’ and G” data for large shear stresses (and higher tank-treading frequencies and larger strains) are meaningful. The cell stiffening behavior that we report in the manuscript (Figure 3c) agrees qualitatively with earlier findings (see for example Kollmannsberger Soft Matter 2011 or Lange BJ 2017) and demonstrates that the method can in principle be used to investigate the non-linear behavior of cells. We added this information to Discussion. Also, we wish to point out that existing microfluidic cell deformability methods (with the exception of a method that we had developed some years ago, see Lange BJ 2015 and Lange BJ 2017) ignore mechanical non-linearites, moreover they mostly ignore frequency-dependent or strain-rate dependent cell behavior, and typically they even ignore dissipate (viscous) properties and only report elastic cell properties. Therefore, we believe that our method represents an advance over these previous methods.

Here you basically say that the measurements you obtain in the high frequency regime do not work. Why are these not excluded, and how do you determine the stress at which we cannot take the G' and G' as unaffected.

We are saying the opposite, namely that measurements obtained in the high frequency regime are valid, but we exclude cells near the channel center that experience a low shear stress and low tank-treading frequency. Regarding the question if there is a shear stress threshold that separates a linear from a non-linear regime, we have investigated this issue in two previous studies (Kollmannsberger2011, Lange2017) and did not find such a threshold. In other words, cells are non-linear, and the complex shear modulus is always affected to some degree by the stress magnitude.

– In Line 580 you say the AFM goes from 0.1 to 150Hz, but the measurements you show are up to 10000 rad/s. What is right?

Thank you, we made a mistake and multiplied the frequency twice with 2 pi. This has now been corrected.